# CORNN: Convex optimization of recurrent neural networks for rapid inference of neural dynamics

**Fatih Dinc**[*]
Department of Applied Physics
Stanford University
Stanford, CA 94305

**Adam Shai**[*]
CNC Program
Stanford University
Stanford, CA 94305

**Mark J. Schnitzer**[†]
Howard Hughes Medical Institute
CNC Program
Stanford University
Stanford, CA 94305

**Hidenori Tanaka**[†]
Physics & Informatics Laboratories, NTT Research, Inc.
Sunnyvale, CA 94085
Center for Brain Science, Harvard University
Cambridge, MA 02138

## Abstract

Advances in optical and electrophysiological recording technologies have made it possible to record the dynamics of thousands of neurons, opening up new possibilities for interpreting and controlling large neural populations in behaving animals. A promising way to extract computational principles from these large datasets is to train data-constrained recurrent neural networks (dRNNs). Performing this training in real-time could open doors for research techniques and medical applications to model and control interventions at single-cell resolution and drive desired forms of animal behavior. However, existing training algorithms for dRNNs are inefficient and have limited scalability, making it a challenge to analyze large neural recordings even in offline scenarios. To address these issues, we introduce a training method termed Convex Optimization of Recurrent Neural Networks (CORNN)[1]. In studies of simulated recordings, CORNN attained training speeds $\sim$100-fold faster than traditional optimization approaches while maintaining or enhancing modeling accuracy. We further validated CORNN on simulations with thousands of cells that performed simple computations such as those of a 3-bit flip-flop or the execution of a timed response. Finally, we showed that CORNN can robustly reproduce network dynamics and underlying attractor structures despite mismatches between generator and inference models, severe subsampling of observed neurons, or mismatches in neural time-scales. Overall, by training dRNNs with millions of parameters in subminute processing times on a standard computer, CORNN constitutes a first step towards real-time network reproduction constrained on large-scale neural recordings and a powerful computational tool for advancing the understanding of neural computation.

## 1 Introduction

Understanding the relationship between neural dynamics and computational function is fundamental to neuroscience research [1]. To infer computational structure from neural population dynamics,

---

[*]These authors contributed equally to this work.

[†]These authors co-supervised this work.

[1]CORNN software and reproduction code are available at https://github.com/schnitzer-lab/CORNN-public

37th Conference on Neural Information Processing Systems (NeurIPS 2023).

neuroscientists regularly collect and analyze large-scale neural recordings using electrophysiological or optical imaging recording methods. Both recording modalities have undergone rapid progress in recent years, yielding a steady increase in the numbers of cells that can be recorded and manipulated [2–7]. Owing to this progress, an urgent need has emerged for developing and refining theoretical and computational tools suited for the analysis of large recording datasets.

Data-driven modeling of neural network dynamics, in which recordings of neural activity are transformed into a computational model through network reconstruction, provides a promising way of bridging experiments and theory [8]. This approach not only complements recent experimental breakthroughs in targeted stimulation of individual neurons ([9, 10]) for hypothesis testing and brain-machine interfaces but also facilitates the distillation of fundamental computational principles from large amounts of experimental data [1, 11, 12]. Historically, especially for recordings of individual or small numbers of cells, hand-crafted network models were used [13–20]. While valuable, traditional hand-crafted models often struggle to handle the large datasets generated in contemporary neuroscience.

To bridge the gap, a new approach has recently emerged in which artificial networks are trained on tasks mimicking those performed in the laboratory by animal models [21–23]. However, this approach lacks the ability to transform observed patterns of neural activity into data-driven, mechanistic computational models. A promising solution to these issues, which retains the advantages of neural network-based modeling, is the training of dRNNs [24–29]. These models, part of a broader suite of computational strategies [11, 21, 30–32], align the dynamics of the neural units within the RNN to the activities of recorded neurons. This data-constrained methodology for RNN training has the potential to modernize estimations of functional connectivity [33]—still a mainstay in systems neuroscience [6]—and provides a viable path to distill underlying computations by extracting the state-space representations of the learned dRNNs [12]. Additionally, dRNNs may open the door to computational control for causal experimentation [9, 10], serving as a valuable computational adjunct to widely used techniques like optogenetics [8, 34].

Despite recent advances in the dRNN framework, how to perform fast and scalable reconstructions of neural activity traces from large-scale empirical recordings has remained unclear. Notably, the slowness of existing dRNN optimization algorithms may often necessitate the use of high-performance clusters and several days of computation in large-scale experiments. These limitations might have been a barrier to the widespread adoption of dRNN approaches. Moreover, to fully harness the potential of dRNNs, it is essential to extend their range of applicability from offline analyses to on-the-fly applications within individual recording sessions. An approach to training dRNNs in a nearly immediate manner would be a key advancement that would enable novel experimental approaches, such as theory-driven real-time interventions targeting individual cells with specific computational or functional roles (Fig. 1). However, realizing these benefits requires having an optimization routine that is fast, robust, and scalable to large networks.

In this work, we present CORNN, a convex solver designed for fast and scalable network reproduction. We demonstrate its accuracy and efficiency compared to alternative network training methods such as FORCE [35] and back-propagation through time (BPTT), which have been employed in previous seminal works utilizing dRNNs [24–29]. Our main contributions include the development of CORNN (Fig. 2), the introduction of an initialization strategy to accelerate convergence (Figs. 2, S4, and S6), and the demonstration of CORNN's scalable speed (Figs. 3, 5, S2, S3, S5, S8, and S9), which enables rapid training on the activity patterns of thousands of neurons (Figs. 4, 5, S7, S8, S9, S10, and S11). CORNN's performance can be further enhanced by 1-2 orders of magnitude through the use of a standard graphical processing units (GPU) on a desktop computer (Figs. S5 and S8). Unlike BPTT and FORCE, CORNN does not require fine-tuning (Figs. S1, S6), making it a user-friendly technology for biologists. Lastly, we highlight CORNN's robustness against non-idealities such as mismatches in the assumed dynamical system equations (Figs. 4, 5, S10, and S11), subsampling of neural populations (Fig. 5), differences in dynamical time-scales (Fig. S10), and existence of non-Gaussian (Fig. S3) or correlated noise (Fig. S11).

By enabling user-friendly, subminute training on standard computers, CORNN represents a first necessary step in transforming data-constrained recurrent neural networks from a theoretical concept into an experimental/computational technology. While this work focuses on introducing and validating the fast solver on simulated benchmarks, future work with CORNN should focus on addressing the challenges that arise when analyzing large-scale neural recordings from experimental neuroscience.

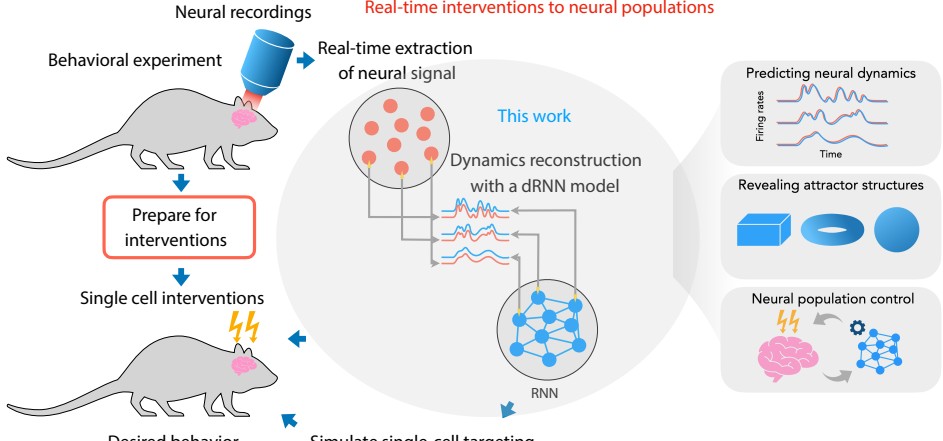

Figure 1: **Using data-constrained recurrent neural networks for the interpretation and manipulation of brain dynamics within a potential experimental pipeline.** This approach centers around online modeling of network dynamics, which can enhance hypothesis testing at the single-cell level and support advancements in brain-machine interface research. The training process is motivated by three objectives: (i) predicting the patterns of neural populations, (ii) revealing inherent attractor structures, and (iii) formulating optimal control strategies for subsequent interventions.

## 2 Approach

### 2.1 Experimental setup for real-time interventions using dRNNs

Previous work suggested the use of dRNNs for real-time feedback between experimental and computational research [8]. To date, however, the use of dRNNs has remained offline. Past studies extracted computational principles from neural recordings by fitting dRNNs *after* data collection [24–29]. Moreover, the use of dRNNs has generally required the expertise of computational neuroscientists, likely due to the complications associated with the optimization and training of neural network models. However, with the advent of large-scale recordings and data-driven approaches, real-time brain-machine interface research is likely a forthcoming application [36–38]. In such use cases, neurobiologists might wish to train dRNNs via a user-friendly approach. Hence, a fast and straightforward dRNN training procedure would enable a new breed of interventional experiments to dissect the brain's microcircuitry.

To appreciate the advantages of fast data-driven reconstruction of neural dynamics, consider a hypothetical scenario in which dRNNs, facilitated by CORNN, enable real-time interventions at the single-cell level (Fig. 1). In this scenario, neural activities from mice performing behavioral tasks are captured and extracted in real-time using advanced imaging or electrophysiological technologies and pre-processing algorithms [2, 3, 7, 39–41]. As the experiment progresses, neural activity traces from each mouse are reproduced by training dRNNs, which are then reverse-engineered to reveal underlying attractor structures. Rapid dRNN training allows for a tight feedback loop between incoming measured data and optimal experimental design, in order to refine the inferred model. Techniques similar to adversarial attacks [42] might be used to devise optimal cell targeting strategies from these dRNNs [8], allowing one to test hypotheses about the computational roles of individual neurons or to identify optimal neurons for use within brain-machine interfaces. Once a perturbation strategy is determined, it can be tested on subsequent trials of the experiment, using the same animal whose recorded neural dynamics led to the trained dRNN. Thus, a fast dRNN training algorithm could allow for better fitting of a dynamical system model to the experimental data and provide a natural testbed to probe hypotheses about the computational structure of the biological neural circuitry.

Motivated by the goals described above, our paper focuses on training dRNNs accurately and as fast as possible. However, experimental concerns and the need for biological interpretability lead to several constraints. Firstly, to ensure real-time communication with the experimental apparatus, we require that the training process takes place on standard lab computers, not clusters. Secondly, given the computational complexity of real-time processing in large scale recordings, especially with thousands of neurons, we expect that at least one of the GPUs is reserved for extracting neural activities

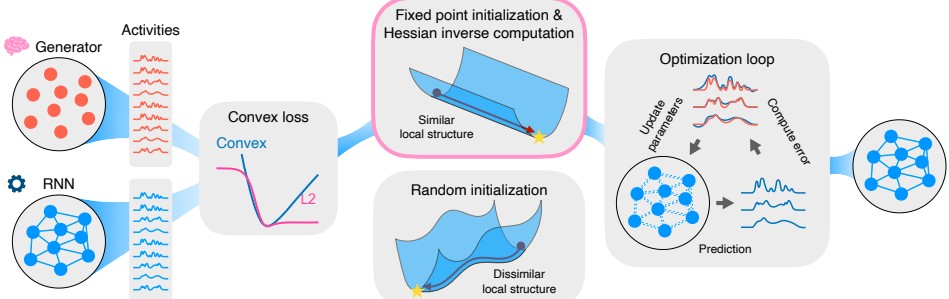

Figure 2: **CORNN: Convex and scalable solver for dRNNs via ADMM [51].** The CORNN algorithm optimizes the parameters of a recurrent neural network so that activity in hidden units align with activities measured from a ground-truth system, we refer to as a generator. The choice of objective function results in a convex loss landscape. The algorithm starts by finding a fixed point for initialization, where the Hessians of all subproblems are aligned to the correlation matrix of neural activities, which can be pre-computed and pre-inverted. The optimization loop then iterates through predicting the neural activities, computing the prediction error, updating the parameter values, and checking for convergence. The final output is the set of optimized parameters, which represents the functional connections between the neurons in the data-constrained recurrent neural network.

from brain-imaging movies [43] (though see [44] for a promising development in smaller scale experiments), or spikes from electrophysiological recordings [7], leaving the central processing unit (CPU), or perhaps a second GPU, for training dRNNs. Finally, maintaining biological interpretability via 1-1 matching of observed and modeled neurons is crucial, as arbitrariness brought by hidden units can explain away existing functional connectivity and/or contributions of observed neurons to underlying attractor structures. These concerns make many traditional machine learning architectures and optimization routines, typically trained on large clusters with billions of parameters, unsuitable for our purposes. This includes recent work on the dynamical system reconstruction paradigm [31, 45–50], which chiefly aims to maximize reconstruction accuracy of dynamical systems without regard to biological interpretability, training speed, and scalability.

## 2.2 Inference model and estimator family of interest

Recurrent neural networks are universal approximators for dynamical systems, whose internal computations can be reverse engineered and interpreted [12, 52]. Thus, we choose leaky firing-rate RNNs as the inference models of CORNN, which follow the dynamical system equations [22]:

$$\tau \frac{\mathrm{d}r_i}{\mathrm{d}t} = -r_i + \tanh(z_i) + \epsilon_i^{\mathrm{con}}, \quad z_i = \sum_{j=1}^{N_{\mathrm{rec}}} W_{ij}^{\mathrm{rec}} r_j + \sum_{j=1}^{N_{\mathrm{in}}} W_{ij}^{\mathrm{in}} u_j + \epsilon_i^{\mathrm{input}}, \tag{1}$$

where $\tau$ is the neural decay time, $z_i$ is the total input to the recurrent unit $i$, $r_i$ is the firing rate of the unit $i$, and $\tanh(.)$ is the pre-defined non-linearity, $W_{ij}^{\mathrm{rec}}$ is the weight matrix for the recurrent connections, $W_{ij}^{\mathrm{in}}$ is the weight matrix from the input to the recurrent units, $u_j$ is the input vector, and $\epsilon_i^{\mathrm{input}}$ is the input noise to the recurrent units. Unlike previous literature, given our goal of reconstruction, we also consider the existence of a conversion noise $\epsilon_i^{\mathrm{con}}$ that accounts for errors and mismatches in the non-linearity. Additionally, the inference model has the constraint $W_{ii}^{\mathrm{rec}} = 0$ preventing any self-excitation of neurons.

Suitable estimators for reconstructing such a model can be obtained (recursively) by discretizing the differential equation:

$$\hat{r}_{t+1,i} = (1-\alpha)\hat{r}_{t,i} + \alpha f(\hat{z}_{t,i}), \tag{2}$$

where we define the time scale parameter $\alpha = \frac{\Delta t}{\tau}$. Here, the model parameters influence $\hat{z}_{i,t}$ explicitly linearly, and implicitly through previous time activities, such that $\hat{z}_{t,i} = \sum_j \hat{x}_{t,j} \hat{\theta}_{ji}$. We define the short-hand notations $x = [r; u]$ and $\hat{\theta} = [W^{\mathrm{rec}}; W^{\mathrm{in}}]^T$ as concatenated matrices. The goal of the estimation problem is to find $\hat{\theta}$ minimizing a loss function.

A reasonable choice for the loss function of this regression problem is the traditional $\mathcal{L}_2$ loss:

$$\mathcal{L}_2(\hat{\theta}) = \sum_{i=1}^{n_{\text{rec}}} \sum_{t=1}^{T} (\hat{r}_{t,i} - r_{t,i})^2. \tag{3}$$

However, due to the time recurrent definition of the general estimator $\hat{r}_{t,i}$, this minimization problem would require backpropagation through time (BPTT). It is not apriori clear if BPTT is the right optimization method for network reproduction, for which one has access to high information content regarding the internal dynamics of the network. We discuss this in the results below.

Instead of using BPTT, we consider a smaller subset of estimators during training where $\hat{r}_{t,i}$ is teacher-forced to $r_{t,i}$ when computing $\hat{r}_{t+1,i}$. This turns the potentially infinite time problem into a single time-step one, which we call "single step prediction error paradigm." Under this condition, the $\mathcal{L}_2$ loss function transforms to a simpler form:

$$\mathcal{L}_2(\hat{\theta}) = \alpha^2 \sum_{i=1}^{n_{\text{rec}}} \sum_{t=1}^{T} (\hat{d}_{t,i} - d_{t,i})^2, \tag{4}$$

where we define $d_{t,i} = \frac{r_{t+1,i} - (1-\alpha)r_{t,i}}{\alpha}$ and correspondingly $\hat{d}_{t,i} = \tanh(\hat{z}_{t,i})$ becomes an estimator consisting of a linear term through a single non-linearity. However, this simplified version of the loss is not convex and leads to a counter-intuitive global loss function with vanishing gradients (Fig. 2).

### 2.3 Convexification of the loss function

Rather than minimizing the naive $\mathcal{L}_2$ loss function, we observe that $\frac{1+\hat{d}_{t,i}}{2}$ follows a linear + sigmoid form. As a result, we replace the $\mathcal{L}_2$ loss function with a cross-entropy loss function that is well known to be convex under this estimator. In fact, in the limit of $d_{t,i} \to \pm 1$, the problem reduces to logistic regression. Instead of a naive replacement, we use a weighted loss of the form:

$$\mathcal{L}_{\text{weighted}}(\hat{\theta}) = \sum_{i=1}^{n_{\text{rec}}} \sum_{t=1}^{T} c_{t,i} \text{CE} \left( \frac{1 + \hat{d}_{t,i}}{2}, \frac{1 + d_{t,i}}{2} \right). \tag{5}$$

Here, we use CE to denote the cross-entropy loss function and $c_{t,i}$ are non-negative weighting factors chosen as $c_{t,i} = [1 - d_{t,i}^2]^{-1}$ for the CORNN solver. The specific choice of $c_{t,i}$, which resembles (but is not) preconditioning [53], is motivated from a theoretical view in Supplementary Section S1.3. The reader can verify that the Hessian of this loss, derived in Eq. (S4), is positive semi-definite. For now, we write down the regularized loss function for CORNN as:

$$\mathcal{L}_{\text{CORNN}}(\hat{\theta}) = \frac{1}{T} \mathcal{L}_{\text{weighted}}(\hat{\theta}) + \frac{\lambda}{2} ||\hat{\theta}||_F^2, \tag{6}$$

where $||\hat{\theta}||_F$ is the Frobenius norm of $\hat{\theta}$ and $\lambda$ is the regularization parameter.

### 2.4 Fundamental principle of CORNN: Subproblem Hessians align to the correlation matrix of neural activity traces

Having convexified the loss function, we next observe that the loss function itself is perfectly separable such that $\mathcal{L}_{\text{CORNN}}(\hat{\theta}) = \sum_i \mathcal{L}_i(\hat{\theta}_{:,i})$, where $\hat{\theta}_{:,i}$ denotes the $i$th column of the parameter matrix $\hat{\theta}$ (See Eq. S1). This means that the original problem can be divided into $n_{\text{rec}}$ sub-problems and solved independently. However, inspired by the fact that solving least-squares for a vector or a matrix target has the same complexity, we can devise a faster method. Specifically, the choice of the specific weighting factors, $c_{t,i} = [1 - d_{t,i}^2]^{-1}$, leads to a shared (approximate) Hessian for all subproblems (See Eq. (S4)):

$$H \approx \frac{1}{T} X^T X + \lambda I, \tag{7}$$

where $X$ stands for $x_{t,i}$ in the matrix form, $I$ is the identity matrix, and $\lambda$ is the regularization parameter. We note that the approximation is exact in the limit $\hat{d}_{t,i} = d_{t,i}$. In other words, we *align the Hessian of each subproblem to the correlation matrix of neural activities.* Unlike for

least-squares minimization, this is a local, not global, Hessian; but as long as we can initialize the network sufficiently close to the optimal solution, we expect that the descent direction with the approximate Hessian converges quickly. In this sense, the approximation in Eq. (7) shows similarities with quasi-Newton methods [53] but is distinct in the pre-computed nature of the Hessian.

Following straightforward algebra, we obtain the gradient as

$$\nabla_{\hat{\theta}} \mathcal{L}_{\text{CORNN}}(\hat{\theta}) = -\frac{1}{T} X^T E + \lambda \hat{\theta}, \tag{8}$$

where we define the prediction error matrix $E_{t,i} = \frac{[d_{t,i} - \hat{d}_{t,i}]}{1 - d_{t,i}^2}$. It is worth noting that each column of the gradient matrix contains the gradient for the scalar subproblem corresponding to the $i$th output. Then, defining the inverse (fixed-point) Hessian matrix as

$$A^+ = \left[ X^T X + T\lambda I \right]^{-1}. \tag{9}$$

we obtain the following update rule (See Eq. (S20))

$$\theta^{k+1} = A^+ X^T X \theta^k + A^+ X^T E^k, \tag{10}$$

where we highlight all quantities that can be pre-computed with blue. In essence, the update rule, which we call "Hessian aligned prediction error (HAPE) update," consists of computing the prediction error followed by matrix multiplications; thus is well suited to be accelerated on a GPU if available, but still can be efficiently computed on CPU cores.

The alignment of subproblem Hessians relies on the assumption that we can find a "close-enough" starting position, called the fixed-point, for the initial parameters $\hat{\theta}^{(0)}$. As we show in Supplementary Section S1.2, a suitable candidate is the approximate least-squares solution:

$$\theta_{\text{ls}} := A^+ X^T Z. \tag{11}$$

Then, the initial set of predictions becomes $\hat{d}_{t,i} := \tanh(\sum_j x_{t,j}(\theta_{\text{ls}})_{j,i})$, which is subsequently refined through the HAPE iterations. This step is computationally negligible, as the blue colored matrix is already pre-computed for the iterations that follow.

In this section, we focused on the leaky firing rate RNNs as described in Eq. (1). However, CORNN can be applied to the other widely used variant of RNNs, *i.e.*, the leaky current RNN described in Eq. (S28) and regularly employed in neuroscience literature [12, 26, 52]. The reproduction of the leaky current RNNs can be made convex by observing the direct link between the firing rates, $r$, and the currents, $z$, via a linear plus non-linear relationship, *i.e.*, $r = \tanh(z)$ and subsequently replacing $d_{t,i}$ with $r_{t,i}$ in Hessian and gradient calculations. To prevent introducing unnecessary complexity, we focus on leaky firing rate RNNs in this work.

## 3 Results

### 3.1 CORNN as a versatile base model

While Hessian alignment, and subsequent HAPE updates, are the core principles driving the fast solver of CORNN, a versatile base model should be able to incorporate various regularization schemes and sparsity constraints. For example, the addition of constraints $W_{ii}^{\text{rec}} = 0$ or a potential low-rank regularization would prevent the Hessian alignment and subsequently the use of fast HAPE updates. However, these constraints could be biologically relevant and perhaps vital to eliminate overfitting in the reproduced network. In fact, for the inference model given in Eq. (1), $r_{t,i} \approx r_{t+1,i}$ for small $\alpha$; making the unconstrained problem ill-posed since self excitation would explain a good amount of the activity of each neuron. Thus, to incorporate the equality constraint and many others, we developed a solver using the alternating direction method of multipliers (ADMM) [51] (summarized in Fig. 2).

In the ADMM solver, the primary problem is divided into two subproblems that are solved individually during subsequent iterations, yet linked through a consensus variable that ensures agreement between the solutions of both subproblems upon convergence. In this framework, the first subproblem solves the unconstrained optimization problem, whereas the second one enforces the constraints/regularization (See Supplementary Section S1 for derivations). This division allows the

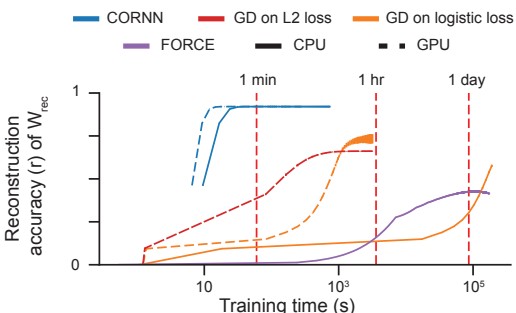

Figure 3: **CORNN reduces training times by several orders of magnitude.** The plots illustrate the relationship between reconstruction accuracy (Pearson's correlation coefficient between ground truth and inferred weights) and training time, measured in seconds on a log scale. The FORCE approach (here, on firing rates) is the default method in neuroscience literature for dRNN training [24, 25, 28, 29]. Parameters: $\alpha = 0.1$, $n_{\text{rec}} = 5000$, $T = 30000$. No input. $\epsilon^{\text{conv}} \sim \text{Poisson}(10^{-3})$, $\epsilon^{\text{input}} \sim \mathcal{N}(0, 10^{-4})$. Lines: median. Error bars: s.d. over 7 networks.

use of fast HAPE updates in the computationally extensive first subproblem, where Hessians can be aligned.

While we only considered the equality constraint in this work, the ADMM framework can be used to add additional L1 and/or nuclear norm regularizations to the learned connectivity matrix [51, 54], and utilize biological priors for further inductive biases. Finally, we note that given the highly irregular nature of the prediction error when $d_{t,i} \approx \pm 1$, we supplied the fast solver with a simple automated outlier detection step explained in Supplementary Section S1.5.

### 3.2 Backpropagation through time on the $\mathcal{L}_2$ loss is suboptimal

In a traditional setting, in which neural activations are hidden and need to be learned through training to provide a correct output, backpropagation through time (BPTT) is necessary to learn long-term dependencies in the data. However, in the network reproduction where neural activations are no longer hidden, it is not clear whether BPTT would be beneficial for the learning, since long-term dependencies are inherently present in the state space of the dynamical system, whose equations are given in Eq. (1). Specifically, given that Eq. (1) has only a single time derivative, the time evolution is performed in a Markovian manner, *i.e.*, given the current state, the next state can be computed without need for additional information (up to a corruption by random noise).

To test whether BPTT is a suitable training algorithm for network reproduction, we examined randomly connected chaotic recurrent neural networks as a synthetic ground-truth benchmark (See Section S3.1 for details). As shown in Fig. S1, contrary to expectation, teacher-forcing enhanced the convergence speed of the BPTT. Moreover, we also observed higher reproduction accuracy with the cross-entropy loss compared to the traditional $\mathcal{L}_2$ loss, inline with our observation from Fig. 2 that $\mathcal{L}_2$ loss leads to vanishing gradients in large error regime. As expected, BPTT took around a minute even in the toy example of Fig. S1 with 100 time points and 100 neurons, approximately two orders of magnitude smaller than a standard calcium imaging session in both dimensions.

### 3.3 CORNN decreases training times several orders of magnitude

Having shown that teacher-forcing can speed-up convergence in chaotic networks, we next validated that chaotic dynamics can be reconstructed via a single step prediction error paradigm that CORNN utilizes. Specifically, we initialized randomly connected recurrent neural networks as generators, extracted neural activities for a fixed duration of length $T$, and trained the dRNNs using CORNN, Newton's solver, gradient descent, and FORCE on varying number of iterations to benchmark their speed and accuracy (See Supplementary Sections S2 and S3.1 for details). All algorithms were able to train, some achieving near perfect accuracies in reproducing the internal connectivity of the generator network (See Figs. 3, S2, and S3). However, FORCE, the current default method in neuroscience literature for dRNN training [24, 25, 28, 29], was 4 orders of magnitude slower (Fig. 3) and needed fine-tuned hyperparameters to barely outperform the fixed-point initialization (Fig. S4).

Whether minimizing the weighted or logistic loss led to more accurate reconstruction depended on the underlying noise distribution (we tested two noise distributions: Poisson(mean) and $\mathcal{N}(\text{mean}, \text{variance})$) of the data as shown in Figs. 3, S2, and S3. However, CORNN outperformed all other algorithms in speed in both cases with several orders of magnitude and converged

in sub-second times even with the CPU implementation, whereas others took up to several days for training (Fig. 3). We also observed increased efficiency for CORNN on a GPU (Fig. S5) and that fixed-point initialization could stabilize other algorithms (Figs. S4 and S6).

## 3.4 CORNN runtimes scale linearly with data size and polynomially with the network size

To this point, we provided validity evidence for the speed and accuracy of CORNN, or single-step prediction error paradigm in general, via the randomly connected chaotic RNN generators. Next, we focused on RNNs with more structured connectivity matrices. Specifically, we trained a model RNN using BPTT to perform a 3-bit flip flop task following [12] as shown in Fig. S7 (See Supplementary Section S3.2 for details). Fig. S7A shows the output of an example test trial, not used for the generator RNN training or the CORNN reproduction, with 200 time points followed by 100 time points of inter trial interval. We observed that CORNN reproduced network performed the task the same way the original network did; potentially making similar mistakes. Thus, as a first step, we confirmed that CORNN can learn to output the same outputs as the original network.

Next, we looked at underlying neural activities of an example (original) generator and reproduced (CORNN learned) network in Fig. S7B. The reproduction matched not only the output of the network, but also the internal firing dynamics of individual units. Yet, as shown in Fig. S7C, the synaptic connections were not perfectly learned even in such a simple scenario with matching models, which required more trials and potentially interventional data [15]. Thus, we next tested whether increasing the number of trials could lead to better reproduction of the original network and whether CORNN can scale well to fit the increased data and network size.

Fig. S8 shows that accurately reproducing the large networks required significantly more trials. Moreover, the data size requirement for the estimation process increased with the number of units in the generator network, which emphasizes the importance of an optimization algorithm that can scale well to multiple day recordings [55]. Fortunately, CORNN had a near-linear scaling of training times *vs.* the increasing number of trials and hence can handle large datasets (See Fig. S8). In the other direction, we observed that the training times scaled polynomially (Fig. S9), following $n_{\text{rec}}^{\beta}$

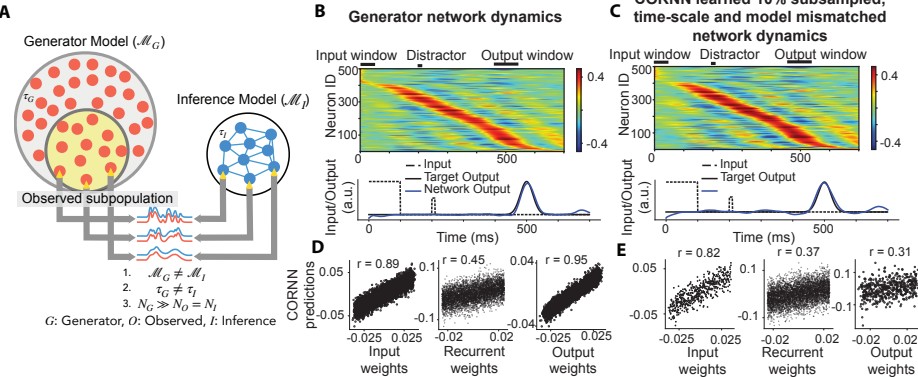

Figure 4: **CORNN can reproduce underlying neural dynamics and attractor structures despite several non-idealities in the timed-response task. A.** To assess the robustness of CORNN against time-scale mismatches, assumed inference models, and unobserved influences from subsampled neural populations, we trained 10 networks using a different generator model than CORNN for the timed-response task. **B.** We conducted a set of novel test trials to evaluate the reproduction accuracy of CORNN when the network was disturbed by a small distractor, which was not present during the learning of the timed-response task. **C.** Despite observing only 500 out of 5000 neurons and with an average 10% mismatch between $\alpha_G$ and $\alpha_I$, the CORNN-learned network reproduced both the neural dynamics and the output. **D.** When reproducing the original network without subsampling or time-scale mismatches but with generator-inference mismatches, the learned weights exhibited imperfect, but non-zero, correlation with the ground truth. **E.** Similar to **D**, we reproduced the network shown in **C** with subsampling and time-scale mismatches. Parameters: $n_G = 5000$, $\epsilon^{\text{RNN}} \sim \mathcal{N}(0, 10^{-2})$ for 100 training trials, one example network, $\alpha_G = \frac{\Delta t}{\tau_G} = 0.1$, and $\Delta t = 1ms$. **B, D**: $n_O = 5000$, $\alpha_I = 0.1$. **C, E**: $n_O = 500$, $\alpha_I \sim \mathcal{N}(0.1, 10^{-4})$.

with $\beta \in [1.3, 2.2]$, *vs.* the increasing number of neurons. Bringing both observations together, we concluded that there is an $O(n_{\text{rec}}^{\beta} T)$ scaling of the empirical training times.

### 3.5 CORNN is robust to a diverse set of non-ideal conditions

To this point, we had considered ideal scenarios in which the generator network shared the same dynamical system equations as the inference network, all neurons were observed, and the time-scales of the generator and inference RNN units matched perfectly. Next, we considered cases in which these assumptions are violated (See Fig. 4A). Specifically, we trained a generator RNN with different governing equations (See Supplementary Section S3.3) to perform a timed-response task (See Fig. 4B), which put the network in a limit-cycle like dynamical attractor state initiated by an input. We tested the existence of the underlying attractor through a novel distractor input, which was not present during the training of the task (See Fig. 4B, C). If CORNN was able to learn the underlying state-space geometry, the reproduced network activities should be able to return to their attractive trajectory under novel perturbations.

When we subsampled the generator networks by 10% and introduced jitter in the time scales, we observed that the inference model reproduced the neural dynamics in the observed population in a novel perturbation trial (See Fig. 4C), even though the sub-connectivity matrix was not well reproduced (See Fig. 4D, E). On the one hand, this observation reinforces the findings of previous literature that the connectivity matrices learned by dRNNs should be interpreted as functional connections, accounting for the behavior of networks at the level of neural dynamics, and not synaptic connections [15, 25]. On the other hand, the ability of the mismatched network to reproduce neural activities in a novel perturbation trial provided evidence that the effects of the underlying attractor structures can be robustly reproduced with CORNN despite severe experimental non-idealities. We quantified these robustness aspects in Figs. 5A,B (subsampling), S10 (time-scale mismatch), and S11 (correlated noise). Moreover, we observed the $n_{\text{rec}}^{\beta}$ scaling for the training times with increased number of network size (Fig. 5C), inline with the results of Fig. S9.

## 4 Discussion

To place in perspective the experimental paradigm enabled by CORNN introduced in this work, we now return to the experimental scenario in Fig. 1. Imagine an experiment in which mice perform a predefined task several times, *e.g.*, for an half hour, with imaging of 3000-4000 neurons. This experimental scenario yields roughly ten million parameters to be trained in the dRNN. The current study showcased CORNN's efficacy in training networks of thousands of neurons, requiring just $O(10)$ iterations, each comparable in complexity to gradient computation and taking seconds. Consequently, training such a network from the initial imaging session would take less than a minute. Once trained, the network can enable real-time planning of experimental interventions, testing

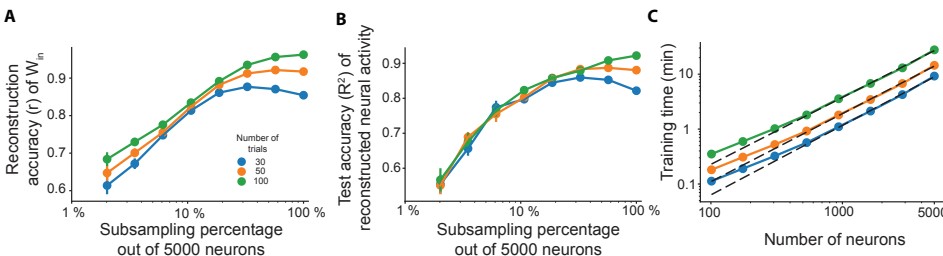

Figure 5: **CORNN runtimes scale polynomially with increasing number of neurons, whereas reconstruction accuracies remain robust to subsampling.** We use the timed-response task as a testbed for quantifying the robustness of CORNN to unobserved influences due to subsampling in neural populations. **A** Reconstruction accuracy of input weights are plotted as a function of subsampling ratio (fraction of observed neurons), **B** reconstruction accuracy of neural activities, measured as the $R^2$ between the ground truth and predicted activations, and **C** the linear scaling of the training times with asymptotic slopes $\in [1.2, 1.3]$ vs increasing number of neurons on a log-log plot, hinting at polynomial scaling. The different colors in the plot correspond to different number of training trials. Parameters: $\alpha = 0.1$, $\epsilon^{\text{RNN}} \sim \mathcal{N}(0, 10^{-2})$ for the training trials. See Supplementary Section S3.3 for further details. Data points: mean, error bars: s.e.m. over 10 networks.

multiple scenarios in parallel to identify several optimal targeting strategies, a template of potential interventions, for neuron combinations driving desired animal behavior. Moreover, with CORNN's rapid convergence and typical inter-trial intervals of a few seconds in behavioral experiments, the network and the interventional strategy can be refined between trials with incoming data streams, facilitating real-time learning.

To understand the timescales for real-time inference with a dRNN for within-trial intervention experiments using calcium imaging, we can look at typical values. Acquiring each image frame takes about 30 ms [2]. While new frames continue to be acquired, motion correction and neural activity extraction can happen in ∼5 ms per frame [56]. In addition to these ongoing imaging steps, the simulation and decision-making with the CORNN-fitted dRNN needs to be performed. The dRNN starts with the current observed activity and can quickly simulate multiple future responses under different input scenarios, *i.e.*, potential interventions. Since the simulation involves only iterative matrix multiplications and point-wise nonlinearities, it finishes very quickly—in just a few ms. For instance, running a 1000neuron RNN forward for 10 timesteps (∼ a second), for 100 different initial conditions, takes $< 6$ ms on a GPU, and $< 25$ ms on a CPU, in our hands. Once a desired intervention is identified from the simulations, the phase mask on a spatial light modulator can be updated in ∼10 ms to optically stimulate the chosen neurons [56]. Putting all the steps together, in under 100 ms one could capture brain activity, simulate future responses, decide on an intervention, and update the optical stimulation parameters. This is fast enough for real-time closed-loop applications. By streamlining network training, CORNN provides dRNNs for these types of experiments.

An important final point in considering the use of CORNN in experimental settings comes from the fact that neural networks are, in general, non-identifiable [57]. That is, for any given settings of the parameters, there are other settings which give the same input-output function. This means that CORNN does not aim to infer the true underlying synaptic connectivity matrix from a neural activity dataset. Instead, the main use of CORNN is to infer an RNN model which recapitulates the dynamical trajectories in a neural population. CORNN may also capture the underlying attractor structure of a system. However, we caution that any claim having to do with attractor structures must be experimentally validated with perturbation experiments that directly test for attractors. In our work, we simulated such experimental validation (Figs. 4 and 5) and found that in the setting tested, CORNN was able to predict the dynamical effects of perturbations on the neural population.

# 5   Conclusion

In this work, we introduced a fast and scalable convex solver (CORNN) for rapid training of data-constrained recurrent neural networks. We showed that CORNN is as accurate yet faster than existing approaches by orders of magnitude and can easily scale to the large datasets of today, training in seconds to a few minutes depending on the data, network size and the availability of a GPU. CORNN, as a base model, lays the groundwork for future developments and can be improved by integrating experimentally relevant assumptions and regularizations within the ADMM framework.

When applied to simulated data from structured networks, CORNN picked up the underlying attractor structure despite non-idealities (Figs. 4, 5, S10, and S11). Inspired by this observation, further development of data-constrained RNNs can support systems neuroscience research aiming to understand inter-area interactions and can complement or augment studies of canonical or pairwise correlations toward characterizations of functional connectivity [33]. Moreover, reverse engineering the learned network may provide a deeper understanding of how neuronal populations contribute to emergent computation, cognition, and memory [12, 30]. Finally, with the advent of interventional methods such as single-cell targeted optogenetics [10], dRNNs can be trained to test causal connections computationally and supply theoretical predictions at a scale previously unachievable for single-cell targeted experimentation [8].

This work constitutes a first step towards the application of CORNN to experimental data. However, several steps remain to apply dRNNs in real-time to interventional experiments. Some example steps may include the transformation of calcium traces or spike trains into traces of firing rates normalized within $[-1, 1]$, applying the CORNN solver developed in this work into first offline and then online experimental scenarios, estimation of neuronal time-scales from the experimental data instead of tuning them as hyperparameters, and perhaps implementing a low-rank regularization approach that opens the door to interpreting the observed dynamics in terms of latent variables [26, 52].

## Acknowledgements

We would like to thank Liam Storan, Udith Haputhanthri, Parth Nobel, Dr. Itamar Landau, Dr. Yoshihisa Yamamoto, Dr. Surya Ganguli, Dr. Jay Mclelland, Dr. John Duchi, and Dr. Stephen Boyd for valuable feedback and insightful discussions. FD receives funding from Stanford University's Mind, Brain, Computation and Technology program, which is supported by the Stanford Wu Tsai Neuroscience Institute. MJS gratefully acknowledges funding from the Simons Collaboration on the Global Brain and the Vannevar Bush Faculty Fellowship Program of the U.S. Department of Defense. FD expresses gratitude for the valuable mentorship he received at PHI Lab during his internship at NTT Research.

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

# S1 Details on CORNN derivations and implementation

## S1.1 Separation into subproblems and derivation of the gradient and Hessian

The loss function given in Eq. (6) can be written in details as:

$$\mathcal{L}(\hat{\theta}) = \sum_{i=1}^{n_{\mathrm{rec}}} \underbrace{\left[ \frac{1}{T} \sum_{t=1}^{T} c_{t,i} \mathrm{CE}\left( \frac{1+\hat{d}_{t,i}}{2}, \frac{1+d_{t,i}}{2} \right) + \sum_{j=1}^{n_{\mathrm{tot}}} \frac{\lambda}{2} \hat{\theta}_{ji}^2 \right]}_{\mathcal{L}_i(\hat{\beta}^{(i)})}, \tag{S1}$$

where $n_{\mathrm{tot}} = n_{\mathrm{rec}} + n_{\mathrm{in}}$. We note that for a given $i$, the functions $\mathcal{L}_i$ depend only on $\hat{\beta}^{(i)} := \hat{\theta}_{:,i}$, i.e., the $i$th column of the $\hat{\theta}$ matrix. In other words, the lack of cross-talk between the columns of $\hat{\theta}$ in the loss function means each output can be regressed independently with respect to the corresponding part of the loss function, i.e., $\mathcal{L}_i(\hat{\theta}_i)$. Thus, for the rest of this section, without loss of generality, we simply considered one instance of the regression, i.e., fix $i$, derived the learning rule, and later generalized to the full problem.

The loss function of interest for the subproblem is:

$$\mathcal{L}(\hat{\beta}) = \frac{1}{T} \sum_{t=1}^{T} c_t \mathrm{CE}\left( \frac{1+\hat{d}_t}{2}, \frac{1+d_t}{2} \right) + \lambda \sum_{j=1}^{n_{\mathrm{tot}}} \frac{(\hat{\beta}_j)^2}{2}. \tag{S2}$$

The gradient can be computed as:

$$\begin{aligned}
(\nabla_{\hat{\beta}} \mathcal{L}(\hat{\beta}))_k &= \frac{1}{T} \sum_{t=1}^{T} c_t \left[ \nabla_{\hat{\beta}} \mathrm{CE}\left( \frac{1+\hat{d}_t}{2}, \frac{1+d_t}{2} \right) \right]_k + \lambda \hat{\beta}_k, \\
&= \frac{1}{T} \sum_{t=1}^{T} \frac{c_t}{2} (1-\hat{d}_t)(1+\hat{d}_t) \left[ \frac{1-d_t}{1-\hat{d}_t} - \frac{1+d_t}{1+\hat{d}_t} \right] x_{t,k} + \lambda \hat{\beta}_k, \\
&= \frac{1}{T} \sum_{t=1}^{T} c_t (\hat{d}_t - d_t) x_{t,k} + \lambda \hat{\beta}_k
\end{aligned} \tag{S3}$$

The Hessian follows directly from the gradient and the fact that $\tanh'(x) = (1 - \tanh^2(x))$:

$$(\nabla_{\hat{\beta}}^2 \mathcal{L}(\hat{\beta}))_{kl} = \frac{1}{T} \sum_{t=1}^{T} c_t (1 - \hat{d}_t^2) x_{t,k} x_{t,l} + \lambda \delta_{kl}. \tag{S4}$$

At this point, it is worth emphasizing that while $x_{t,i}$ are shared across all subproblems, $d_t$ are not. In fact, for any neuron $i$, a different set of $d_{t,i}$ would contribute to the Hessian of the corresponding subproblem. Thus, in its current form, using the exact Hessian would require repeating matrix inversions for each of the $N_{\mathrm{rec}}$ neurons. We performed benchmarking with the exact Newton method, See Supplementary Section S2, though the repeated matrix inversion was no longer feasible for large networks (Fig. 3). To mitigate this, we took a fixed-point solver approach in CORNN, described below.

First, we assumed that one can initialize the full problem with an initial point, call it $\bar{d}_t = \tanh(\hat{\beta}_{\mathrm{fp}}^T x_t)$, such that $\bar{d}_t \approx d_t$. We discussed in Supplementary Section S1.2 on how to estimate $\hat{\beta}_{\mathrm{fp}}$. Then, we replaced the Hessian with a local approximation $\nabla_{\hat{\beta}}^2 \mathcal{L}(\hat{\beta}) \approx \nabla_{\hat{\beta}}^2 \mathcal{L}(\hat{\beta}_{\mathrm{fp}})$ within the vicinity of $\bar{d}_t \approx d_t$, leading to the the local approximate Hessian:

$$(\nabla_{\hat{\beta}}^2 \mathcal{L}(\hat{\beta}_{\mathrm{fp}}))_{ij} = \frac{1}{T} \sum_{t=1}^{T} \underbrace{\frac{1-\hat{d}_t^2}{1-\bar{d}_t^2}}_{\approx 1} x_{t,i} x_{t,j} + \lambda \delta_{ij} \approx \frac{1}{T} X^T X + \lambda I = H_{\mathrm{fp}}. \tag{S5}$$

From this equation, we can infer why the choice of $c_t$ in the main text was scalable. The local Hessian depends only on the inputs, which are shared across all the $n_{\mathrm{rec}}$ independent subproblems, and not on

the predicted or true outputs ($\hat{d}_t$ or $d_t$). Moreover, it needs to be computed once, and at the beginning of the optimization only.

Using the approximate Hessian and gradient, the parameter update rule simply follows:

$$\Delta_{\text{fp}} = -H_{\text{fp}}^{-1}\nabla_{\hat{\beta}}\mathcal{L}(\hat{\beta}). \tag{S6}$$

Even though $\nabla_{\hat{\beta}}\mathcal{L}(\hat{\beta})$ was a vector for the subproblem, given that the (approximate) Hessians for all subproblems are aligned to the correlation matrix, we can promote the gradient to a matrix $\nabla_{\hat{\theta}}\mathcal{L}(\hat{\theta})$ to solve all subproblems at once. Thus, one can *vectorize* the computation for each subproblem, *i.e.*, minimization of $\mathcal{L}_i(\hat{\beta}^{(i)})$ for $i = 1, \ldots, N_{\text{rec}}$, without the need for an external for loop.

At this point, we note that since $H_{\text{fp}} \succeq 0$ is positive semi-definite (PSD), the direction of the parameter updates is guaranteed to be a descent direction. This is because the update direction has a negative angle with the gradient following $\Delta_{\text{fp}}^T\nabla_{\hat{\theta}}\mathcal{L}(\hat{\theta}) = -\nabla_{\hat{\theta}}\mathcal{L}(\hat{\theta})^T H_{\text{fp}}^{-1}\nabla_{\hat{\theta}}\mathcal{L}(\theta) \leq 0$ by positive semi-definiteness. Thus, even if the error between $\bar{d}_t$ and $d_t$ is large for an instance, *e.g.*, fixed-point is incorrectly chosen, it can only hurt the speed of the optimization. Bringing all together, fixed-point updates can be written as:

$$\theta^{(k+1)} := \theta^{(k)} - \gamma_k H_{\text{fp}}^{-1}\nabla_\theta\mathcal{L}(\theta^{(k)}), \tag{S7}$$

where $\gamma_k$ is the user-controlled learning parameter chosen either as 1 (to bypass the line search if fixed-point was close enough) or through back-tracking line search. For this work, we simply picked $\gamma_k = 1$ without any line search as we did not observe any convergence issues. We will simplify Eq. (S7) to its final form when discussing the ADMM solver in Supplementary Section S1.4.

## S1.2   Fixed point initialization

To derive the fixed point update in Eq. (11), we performed a Taylor approximation of the gradient and carried out a single update near the vicinity of the putative fixed-point:

$$
\begin{aligned}
\hat{\beta}^{(t+1)} :&= \hat{\beta}^{(t)} - H_{\text{fp}}^{-1}\nabla_{\hat{\beta}}\mathcal{L}(\hat{\beta}^{(t)}), \\
&\approx \hat{\beta}^{(t)} - \left(\frac{1}{T}X^T X + \lambda I\right)^{-1}\left[\left(\frac{1}{T}X^T X + \lambda I\right)\hat{\beta}^{(t)} - \frac{1}{T}X^T Z\right], \\
&= \hat{\beta}^{(t)} - \hat{\beta}^{(t)} + \left(\frac{1}{T}X^T X + \lambda I\right)^{-1}\frac{1}{T}X^T Z, \\
&= \hat{\beta}_{\text{ls}},
\end{aligned}
\tag{S8}
$$

where a single-step converges to $\hat{\beta}_{\text{ls}} = \left(X^T X + T\lambda I\right)^{-1} X^T Z$, *i.e.*, the solution to the least squares problem on the currents $Z$. Thus, $\hat{\beta}_{\text{ls}}$ is expected to be near to both the solution $\beta^*$ and the regime described by the fixed-point Hessian $H_{\text{fp}}$; thus a suitable candidate to be the fixed-point itself. As an added bonus, this choice did not increase the complexity of the solver; since $\left(X^T X + T\lambda I\right)^{-1} X^T$ has already been pre-computed for the iterative updates.

## S1.3   Interpretation of the weighted convex loss

To interpret CORNN's loss function, we consider the low error limit of the cross entropy $\text{CE}(\hat{p}, p)$ around $\hat{p} \approx p$:

$$
\begin{aligned}
\text{CE}(\hat{p}, p) &= -p\log(\hat{p}) - (1-p)\log(1-\hat{p}), \\
&= -H(p) - \frac{\text{d}}{\text{d}\hat{p}}\left[p\log(\hat{p}) + (1-p)\log(1-\hat{p})\right]\Big|_{\hat{p}=p}(\hat{p}-p) \\
&\quad - \frac{\text{d}^2}{2\,\text{d}\hat{p}^2[t]}\left[p\log(\hat{p}) + (1-p)\log(1-\hat{p})\right]\Big|_{\hat{p}=p}(\hat{p}-p)^2, \\
&= -H(p) + \frac{(\hat{p}-p)^2}{2p(1-p)} + O((\hat{p}-p)^3),
\end{aligned}
$$

where $-H(p)$ is a constant independent of the optimization variables, and thus can be ignored. Then, the low-error limit of the CORNN loss is:

$$\frac{1}{T}\sum_{t=1}^{T} c_t \mathrm{CE}\left(\frac{1+\hat{d}_t}{2}, \frac{1+d_t}{2}\right) \approx \frac{1}{T}\sum_{t=1}^{T}\frac{(d_t - \hat{d}_t)^2}{2(1-d_t^2)^2},$$

$$\approx \frac{1}{T}\sum_{t=1}^{T}\frac{(d_t - d_t - (1-d_t^2)[\hat{z} - f^{-1}(d_t)])^2}{2(1-d_t^2)^2}, \quad \text{(S9)}$$

$$= \frac{1}{T}\sum_{t=1}^{T}\frac{[\hat{z} - f^{-1}(d_t)]^2}{2}.$$

Here, in the second row, we performed a Taylor approximation around the optimal value $\hat{d}_t \approx d_t + (\hat{d}'_t)|_{\hat{d}_t = d_t}(\hat{z} - f^{-1}(d_t))$. In this low error limit, minimizing the scalable weighted loss approximately accounts to minimizing the $\mathcal{L}_2$ loss on the currents. The main difference is, since the original weighted loss is on the firing rates and not currents, the weighted loss can account for the conversion noise, or non-linearity mismatches, that would induce bias in the explicit conversion $z = f^{-1}(d)$. The observation that low-error limit of the CORNN loss approximates the least-squares provided additional support to the claim that least-squares solution serves as a suitable initialization.

### S1.4 Derivation of the ADMM updates

So far, we discussed the fixed-point solver for minimizing the unconstrained CORNN loss with L2 regularization. In this section, we discuss how to add constraints on weights. Specifically, in a general scenario, it is unexpected that all neurons would be connected to each other. Rather, in many cases, one might be interested in introducing a trainable parameter mask such that $\forall (i,j) \in B(i,j)\ W_{ij} = 0$. To account for this scenario, we incorporated our fast solver into the ADMM framework.

#### S1.4.1 Setting up the full problem

In this section, we will first setup the full problem of interest with constraints, and then review the ADMM equations [51]. To start with, we considered a problem of the following form:

$$\text{minimize} \quad \mathcal{L}(\hat{\beta}) = \frac{1}{T}\sum_{t=1}^{T} c_t \mathrm{CE}\left(\frac{1+\hat{d}_t}{2}, \frac{1+d_t}{2}\right) + \lambda \sum_{j=1}^{n_{\text{tot}}}\frac{(\hat{\beta}_j)^2}{2},$$

$$\text{subject to} \quad \forall k \in B(k),\ \hat{\beta}_k = 0. \quad \text{(S10)}$$

We transformed this problem by adding a new variable $\hat{\chi}$, which was constrained to be equal to $\hat{\beta}$:

$$\text{minimize} \quad \frac{1}{T}\sum_{t=1}^{T} c_t \mathrm{CE}\left(\frac{1+\hat{d}_t}{2}, \frac{1+d_t}{2}\right) + \lambda \sum_{j=1}^{n_{\text{tot}}}\frac{(\hat{\beta}_j)^2}{2},$$

$$\text{subject to} \quad \forall k \in B(k),\ \hat{\chi}_k = 0, \quad \hat{\chi} = \hat{\beta}. \quad \text{(S11)}$$

Defining the indicator function $1(x)$ as

$$1(x) = \begin{cases} 0 & \text{if x is true,} \\ \infty & \text{if x is false} \end{cases} \quad \text{(S12)}$$

The ADMM loss function became:

$$\text{minimize} \quad \mathcal{L}_{\text{ADMM}}(\hat{\beta}, \hat{\chi}) = \underbrace{\frac{1}{T}\sum_{t=1}^{T} c_t \mathrm{CE}\left(\frac{1+\hat{d}_t}{2}, \frac{1+d_t}{2}\right) + \lambda \sum_{j=1}^{n_{\text{tot}}}\frac{(\hat{\beta}_j)^2}{2}}_{\mathcal{L}_{\text{CORNN}}(\hat{\beta})} + \underbrace{\sum_{k \in B(k)} 1(\hat{\chi}_k = 0)}_{\mathcal{L}_{\text{constraint}}(\hat{\chi})},$$

$$\text{subject to} \quad \hat{\chi} = \hat{\beta}.$$

$$\text{(S13)}$$

ADMM Lagrangian has two separable loss functions of independent variables, connected through a linear equality constraint, which can be augmented to obtain:

$$\mathcal{L}_\rho(\hat{\beta}, \hat{\chi}, \kappa) = \mathcal{L}_{\text{ADMM}}(\hat{\beta}, \hat{\chi}) + \kappa^T(\hat{\beta} - \chi) + \frac{\rho}{2}||\hat{\beta} - \hat{\chi}||_2^2, \tag{S14}$$

where $\kappa$ is the dual variable. The ADMM steps (after re-defining $\kappa \to (1/\rho)\kappa$) became (See Eqs. (3.5-3.7) in [51]):

$$\hat{\beta}^{k+1} := \underset{\hat{\beta}}{\arg\min}\left(\mathcal{L}_{\text{CORNN}}(\hat{\beta}) + (\rho/2)||\hat{\beta} - \hat{\chi}^k + \kappa^k||_2^2\right), \tag{S15a}$$

$$\hat{\chi}^{k+1} := \underset{\hat{\chi}}{\arg\min}\left(\mathcal{L}_{\text{constraint}}(\hat{\chi}) + (\rho/2)||\hat{\beta}^{k+1} - \hat{\chi} + \kappa^k||_2^2\right), \tag{S15b}$$

$$\kappa^{k+1} := \kappa^k + \hat{\beta}^{k+1} - \hat{\chi}^{k+1}. \tag{S15c}$$

### S1.4.2 Solving first ADMM iteration via fixed-point updates

Similar to the unconstrained case, we started by finding the gradient and the fixed-point Hessian:

$$\nabla_{\hat{\beta}}\left[\mathcal{L}_{\text{CORNN}}(\hat{\beta}) + (\rho/2)||\hat{\beta} - \hat{\chi}^k + \kappa^k||_2^2\right] = -\frac{1}{T}X^T E + (\lambda + \rho)\hat{\beta} + \rho(\kappa^k - \hat{\chi}^k), \tag{S16a}$$

$$\nabla^2_{\hat{\beta}_{\text{fp}}}\left[\mathcal{L}_{\text{CORNN}}(\hat{\beta}) + (\rho/2)||\hat{\beta} - \hat{\chi}^k + \kappa^k||_2^2\right] \approx \frac{1}{T}X^T X + (\lambda + \rho)I, \tag{S16b}$$

where we recall the prediction error matrix:

$$E_{t,i} = \frac{d_{t,i} - \hat{d}_{t,i}}{1 - d_{t,i}^2}. \tag{S17}$$

Then, a single fixed-point update step became:

$$\begin{aligned}
\Delta_{\text{fp}} &= -\left[\frac{1}{T}X^T X + (\lambda + \rho)I\right]^{-1}\left[-\frac{1}{T}X^T E + (\lambda + \rho)\hat{\beta} + \rho(\kappa^k - \hat{\chi}^k)\right], \\
&= \left[X^T X + T(\lambda + \rho)I\right]^{-1}\left[X^T E - T(\lambda + \rho)\hat{\beta} + T\rho(\hat{\chi}^k - \kappa^k)\right], \\
&= A^+ X^T E - (\tilde{\lambda} + \tilde{\rho})A^+\hat{\beta} + \tilde{\rho}A^+(\hat{\chi}^k - \kappa^k),
\end{aligned} \tag{S18}$$

with $\tilde{\lambda} = T\lambda$ and $\tilde{\rho} = T\rho$. Here, we defined the short-hand notation for the inverse matrix:

$$A^+ = \left[X^T X + (\tilde{\lambda} + \tilde{\rho})I\right]^{-1}. \tag{S19}$$

Consequently, the update rule for $\gamma_k = 1$ became:

$$\begin{aligned}
\hat{\beta}^{k+1} &= \hat{\beta}^k + A^+ X^T E^k - (\tilde{\lambda} + \tilde{\rho})A^+\hat{\beta}^k + \tilde{\rho}A^+(\hat{\chi}^k - \kappa^k), \\
&= A^+ X^T X\hat{\beta}^k + A^+ X^T E^k + \tilde{\rho}A^+(\hat{\chi}^k - \kappa^k),
\end{aligned} \tag{S20}$$

where we highlighted all quantities that can be pre-computed with blue. For $\rho = 0$, this reproduces the update rule for the unconstrained problem in Eq. (10). As before, we promoted the vector $\hat{\beta}$ to a matrix $\hat{\theta}$, hence vectorized across the full problem.

### S1.4.3 Solving second ADMM iteration via projection

For this step, the goal is to minimize

$$\text{minimize} \sum_{j=1}^{n_{\text{tot}}} (\rho/2)(\hat{\beta}_j^{k+1} - \hat{\chi}_j + \kappa_j^k)^2, \tag{S21}$$

$$\text{subject to} \quad \forall k \in B(k), \ \hat{\chi}_k = 0.$$

Fortunately, this problem was perfectly separable across the vector entries of $\hat{\chi}_j$. For cases where $\chi_k = 0$ was constrained, this was the only feasible point and thus the answer to that particular problem. For when $j \notin B(j)$, we obtained the minimum loss value by simply picking $\hat{\chi}_j = \hat{\beta}_j^{k+1} + \kappa_j^k$. In short notation, the update rule for this subproblem became:

$$\hat{\chi}^{k+1} := \underset{\hat{\chi}}{\arg\min}\left(\mathcal{L}_{\text{constraint}}(\hat{\chi}) + (\rho/2)||\hat{\beta}^{k+1} - \hat{\chi} + \kappa^k||_2^2\right) = \Pi_B\left(\hat{\beta}^{k+1} + \kappa^k\right), \tag{S22}$$

where we defined $\Pi_B$ as the projection operator to the constraint satisfying subspace (*i.e.*, $\forall k \in B(k), \ \hat{\chi}_k = 0$). Similar to before, we promoted the vector $\hat{\chi}$ to a matrix to vectorize the full problem.

### S1.4.4 Variable initialization and choice of $\rho$

A priori, it was not clear how $\rho$ should be chosen. However, looking at the Hessian inverse $A^+$, we observed that $\rho \geq \lambda$ was an appropriate choice for the step size $\rho$ to have a reasonable impact on the optimization problem. Empirically, we observed that $\rho \gg \lambda$ lead to faster convergence.

We initialized the primal and dual variables with $\chi^0 = \hat{\beta}^0 = \hat{\beta}_{\mathrm{fp}}$, *i.e.* the pre-computed least-squares solution, $\hat{\beta}_{\mathrm{fp}} = A^+ X^T Z$, without additional computational complexity. Moreover, given that ADMM ensured convergence $\hat{\chi} = \hat{\theta}$, the dual variable $\kappa$ was bound to converge to non-zero values only at the diagonal such that the 2nd ADMM step would converge.

To sum up, without any additional computational cost, we defined the initial conditions for all three variables as:

$$\hat{\beta}^0 = A^+ X^T Z, \tag{S23a}$$

$$\hat{\chi}^0 = A^+ X^T Z, \tag{S23b}$$

$$\kappa^0 = 0. \tag{S23c}$$

### S1.5 Automated outlier detection and the pseudo-code for the full solver

Upon close inspection of the prediction error, we observed that the extremely large error values stemming from $d_t \approx 1$ would amplify the conversion noise and carry little-to-no learning signal. Thus, we implemented a simple outlier detection by ignoring errors larger than some value such that $|E_{t,i}| > v \implies E_{t,i} = 0$. In the CPU solver, we scaled the gradient to account for the missing data points. Since these data points constituted small percentages, mostly $< 1\%$ of the full dataset, we did not perform the scaling for the GPU based solver to prevent unnecessary overhead.

It is worth emphasizing that this choice of outlier detection takes away the convexity of the problem since zeroing out part of the prediction error leads to a non-convex global loss. However, the outliers tended to be ignored throughout the optimization process, thus the problem remained convex in remaining data points. Future work can consider a Huber-like [58] global convex robust loss that would automatically detect outliers. For the purpose of this work, we observed that the version of outlier detection we employed here performed well on empirical benchmarks.

For all solvers, we also defined a working precision $\delta$ $(= 10^{-6})$ and whenever $|d_t| \geq |1 - \delta|$, we projected $d_t$ back to $\pm(1 - \delta)$ to prevent numerical overflow errors. For experiments in Figs. S1 and S6 with BPTT, we projected the firing rates $|r_t| \geq |1 - \delta|$ to the boundary $r_t = \pm(1 - \delta)$ for the same reasons. Empirically, this stabilized all solvers and was performed to ensure a fair comparison.

Bringing all together, we state the full CORNN solver in Algorithm 1. We note that for the figures in this work, we simply used a maximum number of iterations (30-100 depending on experiment) instead of monitoring the convergence of CORNN; as CORNN runtimes remained sub-minute and a few more iterations than ideal were perfectly tolerable.

## S2 Implementation details for other solvers

### S2.1 Back-propagation through time

We performed back-propagation through time for the experiments in Figs. S1 and S6. In our experiments, the teacher forcing means that the neural activities at time $t + 1$ is computed via Eq. (2), but we replace $\hat{x}_{t,i}$ in $\hat{z}_{t,i} = \sum_j \hat{\theta}_{t,j} \hat{x}_{t,j}$ with the data $x_{t,i}$, and $\hat{r}_{t,i}$ with $r_{t,i}$. Initially, $\hat{x}_{t,i}$ depends on $\hat{\theta}$ implicitly, so this replacement cuts off the time-dependent computation graph for the gradient. The probability of teacher forcing is the probability with which the replacement is performed at every single time point during the forward propagation of the network. Specifically, we use the following forward-propagation equations:

$$\hat{r}_{t+1,i} = \begin{cases} (1 - \alpha) r_{t,i} + \alpha f(z_{t,i}), & u < p, \\ (1 - \alpha) \hat{r}_{t,i} + \alpha f(\hat{z}_{t,i}), & u \geq p, \end{cases} \tag{S24}$$

---
**Algorithm 1:** Convex and scalable solver for CORNN via ADMM framework
---
  **CORNN.fit** $(X, D, \lambda, \text{nIter} = 30, \rho_{\text{in}} = 100)$;
   % $X$ contains the aggregated input, e.g. firing rates and input units. It is of size $[T \times N_{\text{tot}}]$;
   % $D$ contains the targets. It is of size $[T \times N_{\text{rec}}]$;
   % To start, rescale the regularization parameter $\lambda := T\lambda$
   $\rho := \rho_{\text{in}}\lambda$ %$\rho$ is picked to be comparable to entries of $X^T X$ and rescaled $\lambda$;
   % Pre-compute the inverses once during initialization;
   $A^+ := (X^T X + (\rho + \lambda)I)^{-1}$;
   $X^+ := A^+ X^T$;
   $X^- := A^+ X^T X = X^+ X$;
   $Z := f^{-1}(D)$ %Compute the currents from the targets ;
   $\theta_{\text{fp}} := X^+ Z$%Initialize the fixed-point to the ls solution;
   % Initialize the primary and dual variables;
   $\theta := \theta_{\text{fp}}$ %Initialize to the least-squares solution;
   $\chi := \theta_{\text{fp}}$ %Initialize to the least-squares solution;
   $u := 0 * \theta_{\text{fp}}$ %Initialize the dual variable;
   **for** $i = 1 : \text{nIter}$;
    Predict the targets $\hat{D} = f(X\theta^k)$;
    Compute the prediction error $E^k = (D - \hat{D}) \oslash (1 - D \odot D)$ and perform outlier detection
    % $\oslash/\odot$ stand for element-wise division/product ;
    Update first primal variable $\theta^{k+1} := X^- \theta^k + X^+ E^k + \rho A^+ (\chi^k - \kappa^k)$;
    Update second primal variable $\chi^{k+1} := \Pi_B \left( \theta^{k+1} + u^k \right)$;
    Update the dual variable $u^{k+1} := \kappa^k + \theta^{k+1} - \chi^{k+1}$;
    Probe convergence by checking $||\chi^{k+1} - \theta^{k+1}||$, continue if not converged;
   **end**
  **return** $\chi$
---

where $u$ is a uniform random number generated between $0$ and $1$. It is worth emphasizing that the teacher forcing in this context is distinct from [59], where the teacher signal is acquired not from the real-data but from a second network, though the overall idea is similar.

### S2.2  Current and firing rate based FORCE

We followed the approach by [25] for training the recurrent weights to reproduce target neural activities via FORCE. For both network types, we defined the prediction errors as:

$$e[t] = (\hat{r}[t] - r[t])/\alpha, \quad \text{or} \quad e[t] = \hat{x}[t]) - x[t]. \tag{S25}$$

Here, $\hat{r}[t]$ is the predicted firing rate at time point $t$, vice versa for $\hat{x}[t]$. We no longer performed any teacher forcing, thus $r[t-1]$ and $x[t-1]$ needed to predict $\hat{r}[t]$ came from the previous prediction step of the network, and were not the ground truth values.

The recurrent weights and the corresponding inverse covariance matrix $P[t]$ were updated by the following rule:

$$P[t] = P[t-1] - \frac{P[t-1]\hat{r}[t-1]\hat{r}^T[t-1]P[t-1]}{1 + \hat{r}^T[t-1]P[t-1]\hat{r}[t-1]}, \tag{S26a}$$

$$W_{\text{rec}}[t] = W_{\text{rec}}[t-1] - e[t](P[t]\hat{r}[t-1])^T. \tag{S26b}$$

Here, $e[t]$ was an $N_{\text{rec}}$ dimensional vector, rather than a scalar as was the case for the original work [35]. We initialized $P[t] = \lambda^{-1}I$, where $\lambda$ is the regularization parameter.

### Short-comings of FORCE for dRNN training

Despite its wide use in the literature for dRNN training [24, 25, 27–29], there are several short-comings of this way of training RNNs using FORCE that render it incompatible with the realities of experiments:

1. The training procedure requires a particular range of $\lambda$ to converge (See Fig. S4). Even when FORCE does converge, this is for a very limited range of $\lambda$ and epochs of training.

2. FORCE is initially designed for theoretical investigations and requires the training dataset to be continuous. Even subtle jumps in the data results in stability issues with FORCE training, as the premise of FORCE approach is to have small error from the start to the end [35].

Point 1 necessitates low-level hyperparameter tuning. The only work around to point 2 is to re-initialize the inverse covariance matrix $P$ whenever there is a jump in data points. This practically means that if we want to perform multiple training steps on the same data, each epoch simply corresponds to picking a better initialization for the next epoch, whereas all the learned correlation structure, which is stored in the inverse covariance matrix $P$, needs to be forgotten. The latter point showcases that FORCE is fundamentally ill-suited to many aspects of experimental data, for example, when multiple imaging sessions need to be combined.

### S2.3   Newton's method and gradient descent with single-step prediction error

We implemented Newton's method by using the exact Hessian given in Eq. (S5) for two different scenarios: i) when $c_t = 1$ corresponding to minimizing cross entropy loss, and ii) when $c_t = [1 - d_t^2]^{-1}$ corresponding to the weighted loss. When implementing ii), we performed the outlier detection not only on the gradient, but also the Hessian level; since the division by $[1 - d_t^2]$ could have destabilized the training. Given that each subproblem had a unique Hessian for the Newton descent, we solved the separate subproblems in parallel using multiprocessing tool in numpy.

To implement the gradient descent, we used a single-layer Pytorch model with a linear + tanh structure. We implemented both a CPU and GPU version, and used the ADAM and SGD optimizers. Gradient descent was implemented for two types of loss functions: i) loss function, and ii) cross-entropy with $c_t = 1$.

## S3   Benchmarking details for generator RNNs

### S3.1   Randomly initialized chaotic RNNs

When benchmarking dRNNs on randomly initialized chaotic RNNs, we picked the recurrent weights of the generator randomly from a normal distribution $W_{ij}^{\mathrm{rec}} \sim \mathcal{N}\left(0, \frac{g^2}{n_{\mathrm{rec}}}\right)$ with zero mean and standard deviation of $\frac{g}{\sqrt{n_{\mathrm{rec}}}}$. For $g > 1$ and sufficiently large $n_{\mathrm{rec}}$, these networks can sustain their activities without any external stimulus and exhibit chaotic behavior [60]. For these chaotic generators, we picked $g = 3$ throughout this work to ensure the networks were deep into the sustainable spontaneous activity regime. We chose the chaotic regime specifically, since previous work established that teacher-forcing when training an RNN to output a function can have potentially harmful consequences to the stability of the network training [35]. We aimed to test whether this generalized to the reproduction task we considered in our work.

We note that in figures with randomly initialized chaotic RNNS, where we compared CORNN to other optimizers, we omitted the zero self-excitation constraint, *i.e.*, $W_{ii}^{\mathrm{rec}} = 0$, to remain conservative in our speed comparisons. This is because 2nd order methods (FORCE and Newton descent) cannot incorporate the zero self-excitation constraint without additional optimization steps since projected 2nd order steps are not necessarily descent directions, whereas projected gradient descent is notoriously slower compared to the unconstrained version. In contrast, CORNN was approximately equally fast with or without the ADMM solver; since both ADMM and HAPE updates converged in more or less the same number of iterations.

### S3.2   Training of RNNs for the 3-bit flip flop task

To benchmark CORNN on diverse conditions, on top of the randomly connected RNNs discussed above, we established ground truth data by training RNNs on a 3-bit flip flop task. The architecture of the generator RNN was as follows:

$$r(t) = (1 - \alpha)r(t-1) + \alpha \tanh\left[W_{\mathrm{in}}u(t) + W_{\mathrm{rec}}r(t-1)\right], \tag{S27a}$$
$$o(t) = W_{\mathrm{out}}r(t), \tag{S27b}$$

where $r(t)$ is the hidden state of the RNN at time $t$, $\alpha$ is the time-constant associated with the dynamics, $W_{\text{in}}$ is the weight matrix governing input into the RNN, $u$ is the input into the RNN, $W_{\text{rec}}$ are the recurrent weights, $o$ is the output of the RNN and $W_{\text{out}}$ are the weights which take the hidden state to the output. Since CORNN does not learn $W_{\text{out}}$, to compute the outputs for the CORNN learned networks in Fig. S8, we used the same projection matrix $W_{\text{out}}$ as the original network. See the accompanying code for further details on network training and benchmarking experiments.

### S3.3 Training of RNNs for the timed-response task

To test robustness of CORNN to mismatches between generator and inference network dynamics, we established ground truth data by training leaky current RNNs on a timed-response task with the following architecture:

$$x(t) = (1 - \alpha)x(t-1) + \alpha W_{\text{in}}u(t) + \alpha W_{\text{rec}}r(t-1) + \alpha\epsilon_{\text{RNN}}, \tag{S28a}$$
$$r(t) = \tanh(x(t)), \tag{S28b}$$
$$o(t) = W_{\text{out}}r(t). \tag{S28c}$$

Here, unlike the CORNN and 3-bit flip flop equations (See Eq. (S27)), the time-derivative was at the level of currents ($x$), not firing rates ($r$). Upon receiving an input during $t \in [0, 100]ms$, we trained 10 networks to output a Gaussian pulse centered at $t = 500ms$ with standard deviation $20ms$; otherwise zero everywhere.

During test trials in Figs. 5, S10, and S11, we introduced a novel distractor input ten times stronger than the cue at $t \in [400, 410]ms$. To allow fair comparison between networks trained on different numbers of neurons, we computed the $R^2$ of the neural activities from the first 100 neurons across three 1s trials and performed a uniform average. For the illustrative Fig. 4, to allow visualization of finer details in time-activities after distraction, we instead plotted a scenario with an earlier and weaker (half strength than the cue) distractor at $t \in [200, 210]ms$. To obtain the correlated noise in Fig. S11, we first sampled random noise from a normal distribution, then convolved the i.i.d. noise with a 2D Gaussian kernel across time points (s.d. of $3ms$) and neurons (s.d. of 5 neurons), and re-scaled by the targeted standard deviation. See accompanying code for further details.

### S3.4 The mismatch between leaky-firing and leaky-current RNNs

In this section, we briefly discuss the extent of generator mismatch between leaky-firing and leaky-current networks. To see this, we consider the dynamics of the currents ($z$) under the leaky-firing rate RNNs:

$$\tau\frac{\mathrm{d}z(t)}{\mathrm{d}t} = \tau W^{\text{rec}}\frac{\mathrm{d}r(t)}{\mathrm{d}t} + \tau W^{\text{in}}\frac{\mathrm{d}u(t)}{\mathrm{d}t},$$
$$\tau\frac{\mathrm{d}z(t)}{\mathrm{d}t} = -z(t) + W^{\text{rec}}f(z(t)) + W^{\text{in}}u(t) + \tau W^{\text{in}}\frac{\mathrm{d}u(t)}{\mathrm{d}t}. \tag{S29}$$

Although the current dynamics look similar to the leaky-current RNNs, there are several key differences. Firstly, the term $\tau W^{\text{in}}\frac{\mathrm{d}u(t)}{\mathrm{d}t}$ is added to the leaky-current equation, which can act as an unobserved current when the input stimuli change. Second, the definition of the firing rates are different, with the firing-rate RNNs performing a low-pass filter in time on the linear + non-linear term, $f(z)$, (hence the name, leaky firing-rate RNN). Finally, under non-homogeneous time-scales for the leaky firing-rate RNN, $\tau$ would promote to a diagonal matrix and would not necessarily commute with $W^{\text{rec}}$, so that the Eq. (S29) no longer holds, leading to a stronger mismatch.

## S4 Benchmarking CORNN's speed and scalability

To obtain the CORNN's solver in Algorithm 1, we took five distinct steps:

1. *Single-step prediction error paradigm*, where we teacher-forced $\hat{r}_{t,i}$, the RNN unit activities at time $t$, to the observed neural activities, $r_{t,i}$, to predict $r_{t+1,i}$, with a single-step back-propagation.

2. *Convexification of the loss function*, where we replaced the $\mathcal{L}_2$ loss function with a cross-entropy loss.

3. *Hessian alignment to neural activity correlations*, where we weighted individual samples to align subproblem Hessians to a pre-computable quantity, *i.e.*, the correlation matrix of observed neural activities.

4. *Initialization to a fixed-point*, where we used the approximate least square solution to initialize the networks.

5. *Using ADMM approach* to enforce constraints without increasing algorithmic runtimes.

In this work, we performed several benchmarking experiments to delineate the contribution of each step we take towards the final convex solver to the scalability and speed of CORNN. We proved, with targeted experiments, the need for each of these steps, as we discuss below.

## S4.1 Single-step prediction error paradigm

As discussed in Section 2.2, the first step we took was to restrict the family of interest for the inference models to the models that can perform single-step predictions well. This simplifying assumption mapped the recurrent estimation problem to a single layer feed-forward network with a sigmoid/tanh non-linearity. Here, the inputs of the network are $u_{t,i}$ and $r_{t,i}$, whereas the outputs are $r_{t+1,i}$. Since this step eliminated the need for the back-propagation through time, both the training times and memory requirements decreased significantly (Compare Figs. S1 and S2).

However, a priori, it was not clear whether the restrictive assumption of single-step prediction error minimization would lead to decreased performance. To test this, we performed the experiments in Fig. S1 and observed that teacher forcing increases the efficiency of the training, and was not deleterious. A potential explanation may lie in the observation that, unlike the traditional scenario where a sparse error signal needs to be propagated back in time to perform credit assignment, network reproduction training had access to rich variety of error signals at each time step and for each neuron.

## S4.2 Convexification of the loss function

As discussed in Section 2.3, the convexification of the loss function plays an important role in speeding up the learning process. However, by itself alone, convexification is not the reason of the speed improvement. For example, in Fig. 3, gradient descent on $\mathcal{L}_2$ loss converges faster than the gradient descent on the logistic loss. On the other hand, for smaller networks with 200 neurons (Figs. S2, S3, S5), convex loss function merits the use of Newton descent and leads to faster convergence than gradient descent. Yet, as shown in Figs. S2 and S5, CORNN is at least an order of magnitude faster even for these small networks. Hence, though convexification of the loss function is crucial, it cannot explain the speed or the scalability of CORNN by itself.

## S4.3 Hessian alignment

The main contribution of our work is to pick $c_{t,i}$, which approximates the problem as a local least-squares as discussed in Supplementary Section S1.3, and aligns the Hessians of each subproblem to a pre-computable quantity, *i.e.*, the correlation matrix of neural activities, as discussed in Section S1. This alignment allows the use of HAPE updates that have complexity of gradient descent but converge in $O(10)$ iterations thanks to the approximate Hessian. Figs. S2 and S5 show that Newton's descent with exact Hessian computations on the weighted CORNN loss is an order of magnitude slower than CORNN. Hence, the Hessian alignment was a cruical step to speed up the learning, but most importantly allow scaling to large networks due to its gradient-level complexity.

## S4.4 Fixed-point initialization

The fixed-point initialization relies on an approximate solution when the final error is low (See Supplementary Sections S1.2 and S1.3). This initialization is a good first guess, since, if there was no conversion noise at all, one could have simply performed least-squares on the currents estimated via $z = \mathrm{arctanh}(r)$. This is akin to regressing to the inverse of the output before the non-linearity in a single layer network. Though such an approach can provide a good first guess, it is not robust to conversion noise. This was shown in Figs. 3 and S3, where non-Gaussian noise components lead to low accuracy for the initialization. Finally, the least-squares solution does not satisfy any additional

constraints on the connectivity matrix, thus is not suitable as a valid solution itself. Yet, since the least-squares solution is computationally negligible, we used it as an initialization for CORNN.

### S4.5 Alternating direction method of multipliers

We chose to use ADMM approach to enforce the no self-excitation constraints and present a versatile base model that can incorporate additional design choices such as L1 and/or nuclear norm regularization. In our comparison experiments, we omitted any constraints since other solvers were considerably slower, or outright incompatible. However, in our speed benchmarking results in Figs. 5, S8, and S9, we used the full CORNN solver with the ADMM described in Algorithm 1. The choice of ADMM for enforcing constraints is rooted in our observation that both the ADMM steps and the original HAPE updates take $O(10)$ iterations to converge. Moreover, other parts of the ADMM updates are negligible compared to the primal problem, where the HAPE updates take place. In other words, the fact that HAPE updates and ADMM converge in similar steps creates a mutual synergy that increases accuracy and promotes versatility in our work without hurting convergence times.

## S5 Reproducibility

We provided the code to reproduce each figure in our Github. All experiments, except for those in Figures 3, 5, and S4, were run on a standard computer with Geforce RTX 3090 Ti GPU and Intel Core i9-9900X Skylake X 10-Core processor. Experiments in Fig. 5 were run on a computer with two Geforce RTX 3080 Ti GPUs (though using only one) and Intel Core i9-10980XE 18-Core processor. Experiments in Figure S4 were divided to run on both of the aforementioned computers. Experiments in Figure 3 were run on a computer with two NVIDIA Geforce RTX 4090 (though using only one) and an AMD Ryzen Threadripper PRO 5975WX 32-Core processor (though using only 5 cores due to memory limitations). Illustrative figures are plotted on an Apple Macbook Air with M1 Chip. See the accompanying code provided in the Github repo[2] for the specific hyperparameters used for the experiments.

---

[2]Reproduction code is available at https://github.com/schnitzer-lab/CORNN-public

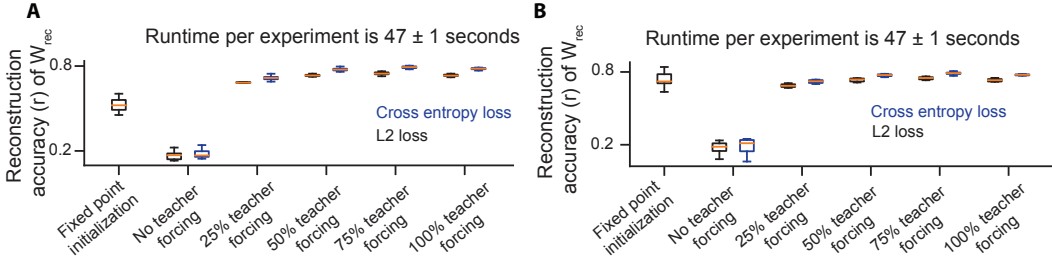

Figure S1: **Backpropagation through time (BPTT) on an $\mathcal{L}_2$ loss function is suboptimal, whereas teacher-forcing can increase the reconstruction accuracy.** To assess the optimality of BPTT on an $\mathcal{L}_2$ loss function and investigate the potential drawbacks of teacher-forcing, we conducted reproduction experiments on chaotic networks with low (**A**) and high (**B**) input to conversion noise ratio. We varied the levels of teacher forcing applied to BPTT and employed two distinct loss functions: cross-entropy and $\mathcal{L}_2$ loss function. Surprisingly, our results challenge traditional beliefs, since cross-entropy with teacher forcing emerged as the superior method compared to BPTT. This superiority may be attributed to the rich information content available during network reproduction training. Parameters: **A, B**: $\alpha = 0.9$, $n_{rec} = 100$, $T = 100$, $n_{epoch} = 3000$. $\forall t = 1, \ldots, T$, $u(t) = 0.1$ with $W^{in} \sim \mathcal{N}(0, g^2)$ and $g = 3$. **A**: $\epsilon^{conv} \sim \mathcal{N}(0, 10^{-8})$, $\epsilon^{input} \sim \mathcal{N}(0, 10^{-4})$. **B**: $\epsilon^{conv} \sim \mathcal{N}(0, 10^{-10})$, $\epsilon^{input} \sim \mathcal{N}(0, 10^{-2})$. Box plots are obtained from 10 random generator networks shared across optimization algorithms. Orange lines denote median values; boxes span the 25th to 75th percentiles; outliers are not shown.

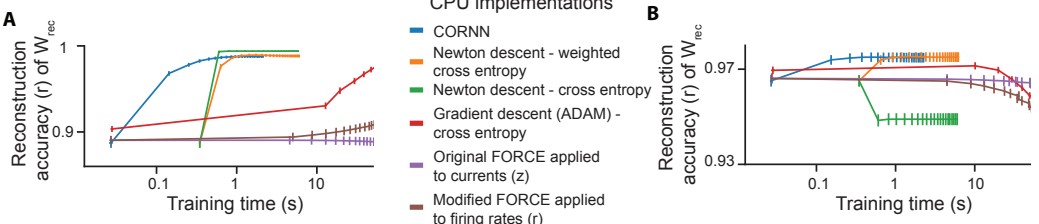

Figure S2: **CORNN achieves a better combination of speed and accuracy compared to alternatives, dramatically accelerating convergence.** The plots illustrate the relationship between accuracy (Pearson's correlation coefficient between ground truth and inferred weights) and training time, measured in seconds on a log scale, with low (**A**) and high (**B**) input to conversion noise ratio. The blue line represents the performance of our newly developed method, which combines fixed-point initialization and CORNN, and achieves high accuracy while significantly accelerating convergence compared to other methods. Parameters: **A-B**: $\alpha = 0.1$, $n_{rec} = 200$, $T = 3000$. No input. **A**: $\epsilon^{conv} \sim \mathcal{N}(0, 10^{-8})$, $\epsilon^{input} \sim \mathcal{N}(0, 10^{-4})$. **B**: $\epsilon^{conv} \sim \mathcal{N}(0, 10^{-10})$, $\epsilon^{input} \sim \mathcal{N}(0, 10^{-2})$. In all plots, data points indicate median values, and error bars denote standard error of the mean (SEM) over 100 runs of the simulation.

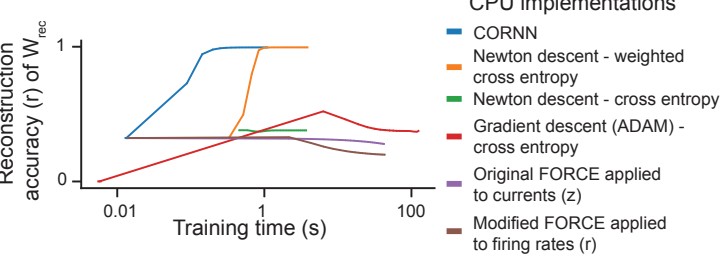

Figure S3: **CORNN is robust to non-Gaussian noise injections.** Same as in Figure S2, but with parameters: $\alpha = 0.1$, $n_{rec} = 200$, $T = 1500$, No input. $\epsilon^{conv} \sim$ Poisson$(10^{-1})$, $\epsilon^{input} \sim$ Poisson$(10^{-2})$.

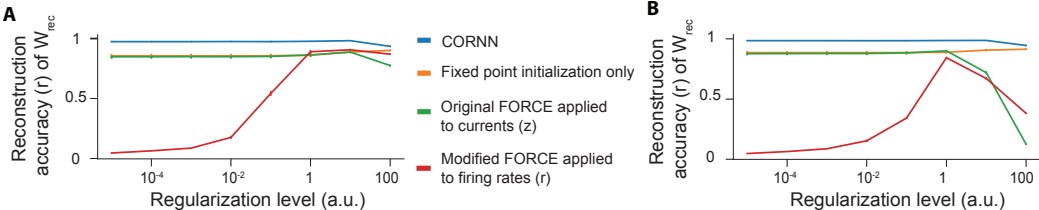

Figure S4: **The traditional FORCE approach is severely suboptimal, underperforms compared to fixed point initialization, and is hyper-sensitive to the regularization level.** This plot describes the relationship between the reconstruction accuracy (correlation between ground truth and inferred weights) and the strength of regularization applied to the weights during training. The results show that while the FORCE algorithm, as previously applied to a similar task [24, 25, 27–29], is highly sensitive to the choice of hyperparameters, CORNN, whose accuracy is practically bounded below by the initialization, is not. Moreover, in majority of the cases, FORCE cannot train beyond the fixed-point initialization we developed in this work. **A** with fixed point initialization and **B** with random initialization. Parameters are the same as in Fig. S2**A**. In all plots, data points indicate mean values, and error bars denote SEM over 10 runs of the simulation.

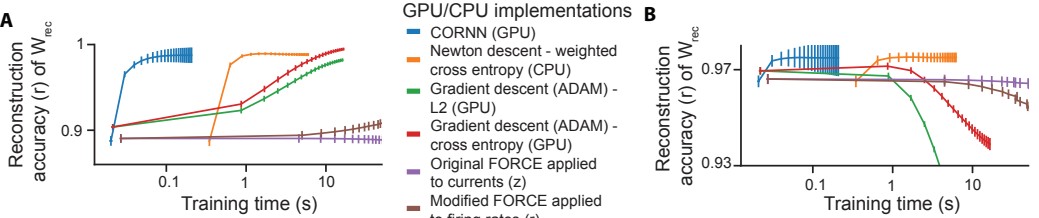

Figure S5: **CORNN can be drammatically accelerated with on a standard GPU.** The same as in Fig. S2, but with GPU acceleration for CORNN and gradient descent algorithms.

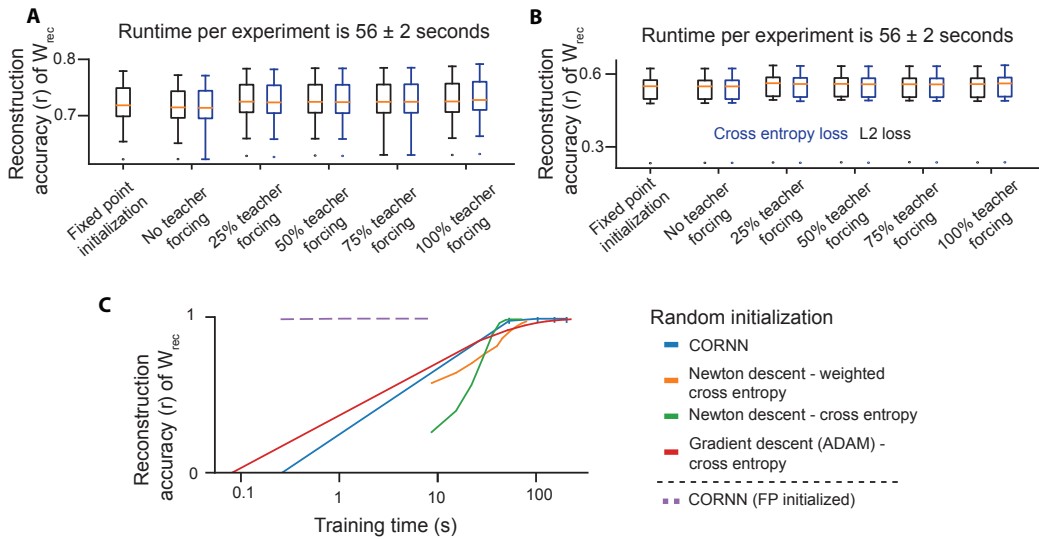

Figure S6: **Fixed-point initialization speeds up the convergence. A-B** Same as in Figure S1, but with fixed-point initialization. **C** We conducted an ablation study to investigate the effectiveness of our proposed fixed-point initialization method. The results show that the CORNN algorithm with fixed-point initialization (purple) outperforms all other methods, including the vanilla CORNN without this fixed-point initialization technique. Parameters: $\alpha = 0.1$, $n_{\text{rec}} = 500$, $T = 10000$, $\epsilon^{\text{conv}} \sim \mathcal{N}(0, 10^{-10})$, $\epsilon^{\text{input}} \sim \mathcal{N}(0, 10^{-2})$. All algorithms are ran on CPU. For **A,B**, the boxes cover from the lower quartile to the upper quartile, with an orange line at the median. For **C**, in all plots, data points indicate median values, and error bars denote SEM over 100 runs of the simulation.

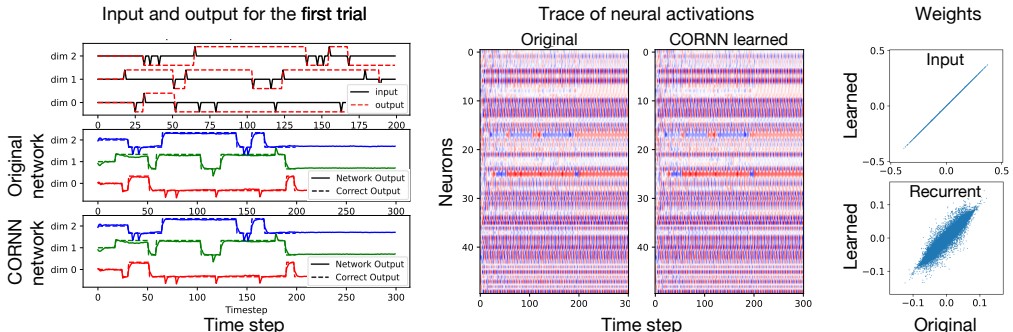

Figure S7: **CORNN can reproduce an RNN performing 3-bit flip flop task through dRNN training.** We use CORNN to reproduce the behavior of a synthetic RNN trained on a 3-bit flip-flop task [12], where the network is given a short input pulse at a random time. If the input pulse is +1, the output of the network must transition to +1 or stay at +1. If the input pulse is -1, the network's output must transition to -1 or stay at -1, ignoring input pulses that do not match. Through dRNN training, CORNN is able to accurately reproduce, *left*, the inputs and outputs of the task-trained RNN, *middle*, spatiotemporal pattern of the neural population dynamics, and, *right*, the inputs and (partially) recurrent weights of the synthetic RNN. Parameters: $\alpha = 0.9$, $n_{\text{rec}} = 1000$, $T = 200$ trials (100 data points per trial), $\epsilon^{\text{conv}} \sim \mathcal{N}(0, 10^{-6})$, $\epsilon^{\text{input}} \sim \mathcal{N}(0, 10^{-4})$.

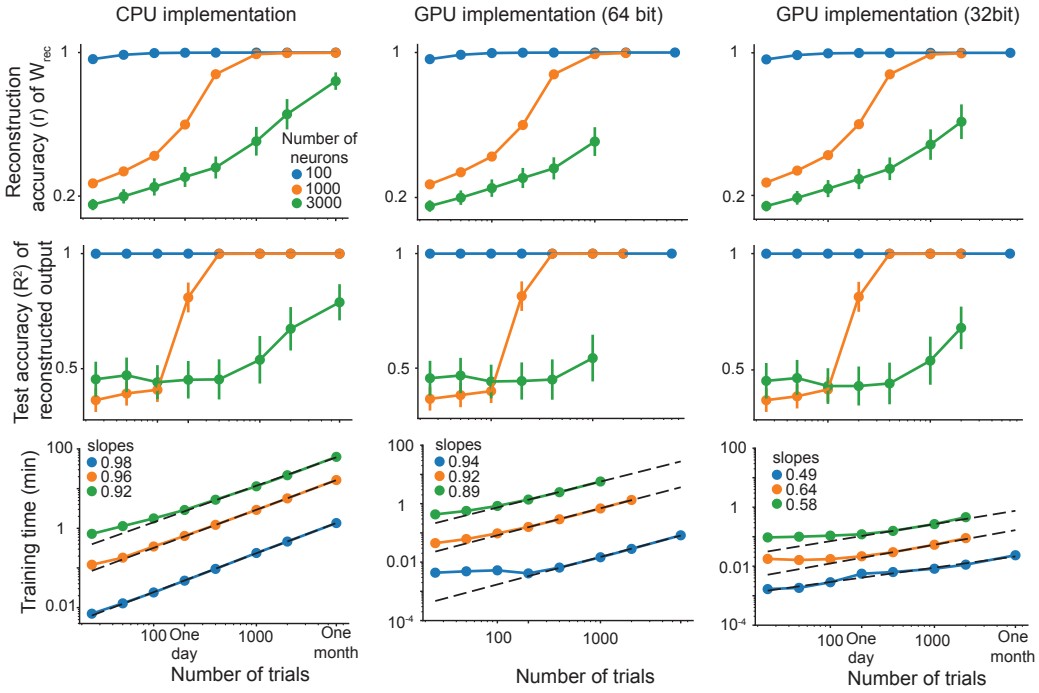

Figure S8: **CORNN runtimes scale linearly with increasing trial data, which can facilitate learning from larger networks.** *Top*, reconstruction accuracy of network weights, measured as the correlation between recurrent weights, are plotted as a function of dataset size (number of trials), *middle*, reconstruction accuracy of network outputs, measured as the $R^2$ between the output units, and, *bottom*, the linear scaling of algorithmic wallclock runtimes with near-unity slope vs increasing number of trials on a log-log plot. The different colors in the plot correspond to different number of neurons in the generator network. Missing data points correspond to the limits of the GPU memory, which can be mitigated by computing prediction errors in temporal chunks in real-data applications, though not implemented for this case study. Parameters: $\alpha = 0.5$, $\epsilon^{\text{conv}} \sim \mathcal{N}(0, 10^{-6})$, $\epsilon^{\text{input}} \sim \mathcal{N}(0, 10^{-4})$. In all plots, data points indicate mean values, and error bars denote SEM over 20 runs of the simulation.

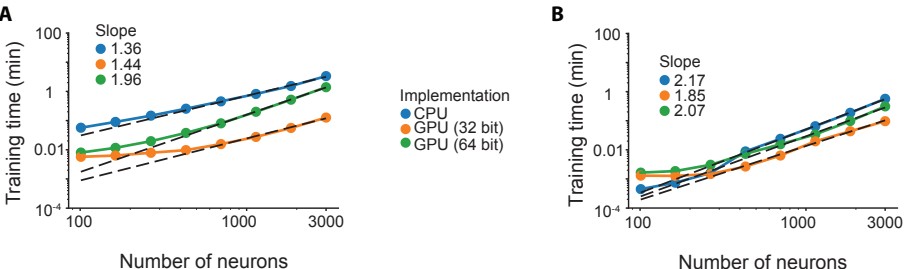

Figure S9: **CORNN scales polynomially to large network sizes.** Using the networks with 3000 neurons trained for the 3-bit flip flop task in Figure S7, we subsampled neural populations and collected CORNN runtimes for CPU and GPU implementations. Parameters: $\alpha = 0.5$, $\epsilon^{\text{conv}} \sim \mathcal{N}(0, 10^{-6})$, $\epsilon^{\text{input}} \sim \mathcal{N}(0, 10^{-4})$. **A** with 200 trials (20,000 data points) and **B** with a single trial (100 data points).

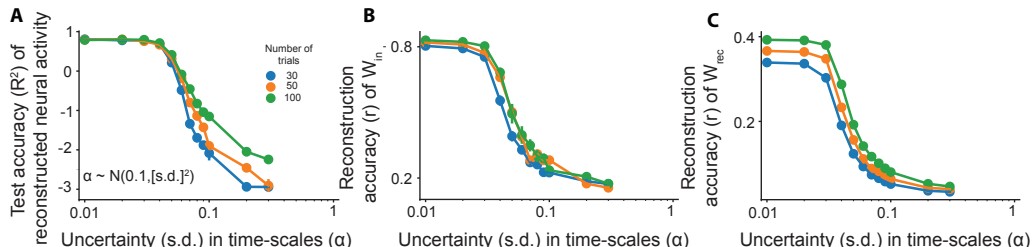

Figure S10: **CORNN is robust to variations in the time-scales compared to the ground truth.** Same as in Figure 5, but with 500 observed neurons and varying levels of uncertainties in time-scales.

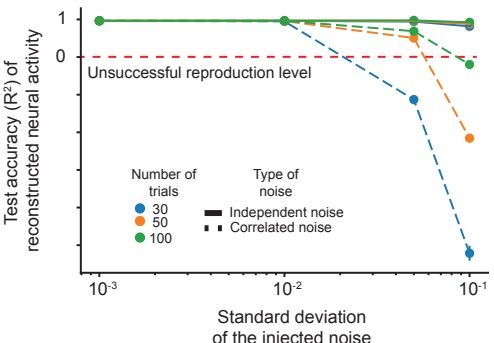

Figure S11: **Correlated noise presents a challenge for CORNN, which can be mitigated by increasing the data size.** Same as in Figure 5, but with the fully observed network and varying levels of injected random and correlated noise (for details, see Section S3.3).

