# OpenReview forum: "CORNN: Convex optimization of recurrent neural networks for rapid inference of neural dynamics"
_NeurIPS.cc/2023/Conference — NeurIPS 2023 poster_

### Official Review · Reviewer_mPZo · 2023-06-27

**Soundness:** 3 good
**Presentation:** 3 good
**Contribution:** 3 good
**Rating:** 7
**Confidence:** 5

**Summary:**

The motivation behind this paper is to replace FORCE/RLS fitting of biologically-flavored RNNs to data by a faster and more robust alternative training scheme. This is accomplished by 1) altering the training objective to a cross-entropy loss; 2) using an approximate pre-computable Hessian to facilitate the use of Newton steps; 3) use of a suitable fixed point initialization; 4) using teacher forcing (training only one-step losses); 5) employing ADMM to enforce constraints (e.g., no self-excitation). Experiments show improvements to various metrics, including accuracy of recovered weights, fits to observed activity, and runtime.

The stated goal of the project is to produce a method fast enough for real-time analysis, though this work focuses only on weight inference and fitting accuracy of the firing rates. Still these models are increasingly used in the field to fit neural data, so better algorithms for doing so are needed.

**Strengths:**

- Good algorithm performance as assessed along multiple dimensions (speed, inference accuracy, reconstruction accuracy).
- Thoughtful comparisons to other baseline models.
- Emphasis on real-time and streaming settings.
- Experiments targeting model mismatch.
- ADMM approach allows for many different types of potential constraints.

**Weaknesses:**

- Addresses some key sources of potential model error but not others (e.g., correlated noise, wrong nonlinearity, mismatch between network timescale and measurement timescale for some data types).
- No experiments on real data. Even if accuracy of weight recovery is impossible to assess, reconstruction accuracy would be.

**Questions:**

- It is unclear to me what the difference is between panels A and B of Figure 3. Just different parameters? It would help if this were reflected in the titles.
- Figure 4: I realize that the authors would like to zoom in to show differences between models, but the choice of y axis range obscures the fact that these are modest to minor differences between the models.
- The authors use correlation between the true and inferred weights as their accuracy measure, but this (so far as I can see) first noted in the caption to Figure 4, even though it occurs in Figure 3.

**Limitations:**

1. Equation (1) is not the usual form for the equations defining these networks (e.g., refs 22, 24). Typically, the leak term is on the voltage variable, not the firing rate, and the nonlinear transfer is from voltage to firing rate, not input current to effective drive. It looks to me as if this choice is essential to make the cross-entropy convex in $\theta$. If so, this should be discussed, since it's a qualitatively minor change that has large implications for the feasibility of the approach.

2. As noted above, it seems to me there are five distinct technical pieces, and it is not always clear what is doing the heavy lifting. For instance, Figure 3 (and S3) seems to indicate that initialization in these models is doing a huge amount of work, which is unsurprising but not well emphasized in the text. Better delineation in the main text of which choices are most important to the results would be helpful.

---

> ### Author Rebuttal · Authors · 2023-08-08
>
> We thank the Reviewer for their expert review, careful reading, and identification of typos, which we have now fixed. We especially appreciated the expert parsing of our approach into 5 distinct components; this showcased how well the Reviewer understood and appreciated our research. With the Reviewer’s permission, we would like to use similar language to describe the contributions of each individual component of the fast CORNN solver, within a new supplementary section that we plan to add. Our detailed responses to each question are below.
>
> - (Request for more simulations and real data applications)
>
> To address the request by the Reviewer for incorporating more non-ideal conditions, we performed new experiments with correlated (Fig N1) and non-Gaussian noise (Fig. N2-3). The reviewer also raised a need to check non-linearity mismatch. We believe the current manuscript addresses these points in two ways. In particular, we studied:
>
> i) effect of conversion noise, which creates an inevitable mismatch between non-linearities. Instead of tanh(.) non-linearity, the conversion noise uses a time and neuron dependent non-linearity of the form “tanh(.) + \epsilon.”
>
> ii) a leaky-current RNN in Fig. 7, whose equations do not match those of the leaky firing rate RNNs. We presume that using a different data generator is more stringent than having a different non-linearity on the same model.
>
> Yet, we agree that processing datasets where firing rates are not bounded within $[-1,1]$, which might stem from other non-linearities, will require some form of data transformation. This is part of the future work that needs to be addressed to apply CORNN to real data, though previous work successfully addressed similar concerns before (See Rajan et al. 2016, Perich et al. 2021). Please also see our response to the final question of reviewer 1Kip, for a related discussion of real data applications and how we changed the manuscript accordingly.
>
> - Figure 4: (...) minor differences between the model accuracies.
>
> We thank the Reviewer for this comment. Indeed, our primary claim about CORNN is that it enables faster training, rather than increased accuracy. On the other hand, with the revised Fig. 4 (See Fig. N2 in the shared PDF), the differences in accuracy are far more clear between models. This is partially because we tested the optimizers on a large-scale scenario with 5000 neurons, as opposed to a few hundred neurons in the prior Fig. 4.
>
> - Equation (1) is not the usual form (...)
>
> Both leaky firing rate and leaky current RNNs are regularly used in the field of neuroscience. After receiving the reviews we realized that our arbitrary choice to focus on one architecture made it seem like our approach was not applicable to the other. However, CORNN can be trivially implemented for both types of RNNs. To address the concern, we added the following paragraph to the end of the Approach section:
>
> “In this section, we focused on the leaky firing rate RNNs as described in Eq. (1). However, CORNN can be applied to the other widely used variant of RNNs, i.e., the leaky current RNN described in Eq. (S27) and regularly employed in neuroscience literature [citation]. The reproduction of the leaky current RNNs can be made convex by observing the direct link between the firing rates, $r$, and the currents, $z$, via a linear plus non-linear relationship, \emph{i.e.}, $r = \tanh(z)$. To prevent introducing unnecessary complexity, we focus on leaky firing rate RNNs in this work, though the generalization to leaky current RNNs can be trivially performed by replacing $d_{t,i}$ with $r_{t,i}$ in Hessian and gradient calculations.”
>
> - Better delineation (...) of which choices are most important (...)
>
> We thank the Reviewer for this extremely helpful comment. The Reviewer is right, we should have made this point clear. Moreover, this comment made us realize that we dissected all aspects but point 3) regarding the effect of initialization in our experiments. To address this, we started a new large-scale experiment with non-Gaussian noise (Fig. N2) and added a new experiment with Poisson noise where the initial guess with least-squares is quite suboptimal (Fig. N3).
>
> To discuss the effect of each step in our convexification procedure, we plan to add a new section to the supplementary, breaking down the steps as suggested by the Reviewer and explaining how they are tested throughout the paper. Here are bullet points describing how we tested each of the following aspects:
>
> 1) “altering the training objective to a cross-entropy loss:” In Fig. 4, and the new Fig. N2 that will replace Fig. 4, we are testing the convergence accuracy and speed of algorithms trained on L2 loss vs cross-entropy loss vs weighted cross-entropy (CORNN) loss.
>
> 2) “using an approximate pre-computable Hessian to facilitate the use of Newton steps:” In Fig. 4, which will now become a supplementary figure, we tested Newton’s descent with exact Hessian computations on the weighted CORNN loss.
>
> 3) “use of a suitable fixed point initialization:” Our new experiments (Figs. N2-N3) will address this point. Though, as noted in the main text, the least-squares solution used for the initialization does not satisfy any additional constraints on the connectivity matrix, thus is not suitable as a valid solution itself.
>
> 4) “using teacher forcing (training only one-step losses):” Teacher forcing is at the heart of our work, without which training a network with 10 million parameters under a minute would not have been possible. Fig. 3 shows that BPTT without teacher-forcing is not necessarily better than teacher-forced approach, whereas (old) Fig. 4 shows that FORCE without teacher-forcing does not necessarily have better accuracy.
>
> 5) “employing ADMM to enforce constraints (e.g., no self-excitation):” ADMM approach is not related to the speed of the solver, rather the ability to incorporate additional constraints, though this comes with no added complexity.

---

> > ### Comment · Reviewer_mPZo · 2023-08-10
> >
> > I appreciate the authors' thorough and thoughtful replies to all reviewers' comments. I believe this work will prove a valuable contribution to those performing model fitting under real-time constraints in neuroscience experiments.

---

> > > ### Comment · Reviewer_mPZo · 2023-08-14
> > >
> > > I am satisfied with the authors' response to my technical queriers. I agree that the lack of application to a real dataset is a deficit, and I share Reviewer GD24's concerns about identifiability, but I am raising my score, as it seems likely this work will be of benefit to those interesting in real-time model fitting.

---

> > > > ### Author Response · Authors · 2023-08-15
> > > > **Response to the official comment**
> > > >
> > > > Thank you very much for your response and consideration. We truly appreciated your comments and suggestions. As for the remaining concerns, please see our final comments to the reviewer GD24. We hope that we were able to address both concerns substantially there.

---

### Official Review · Reviewer_Gp7K · 2023-07-03

**Soundness:** 4 excellent
**Presentation:** 3 good
**Contribution:** 4 excellent
**Rating:** 8
**Confidence:** 4

**Summary:**

The paper introduces a new method for training RNNs that allows fast convergence which could make realtime parameter update possible.

**Strengths:**

- The method conceptually is simple and intuitive.
- It tackles an important problem which is the time consuming nature and computing intensivity of ML models
- It achieves impressive results through simple hacks
- One possible use of this is also quick experimentation with RNN models
- Outlier detection is included

**Weaknesses:**

- The methods section is easy to understand until the end of setion 2.3 where there are many equations with variables that are not defined either in the main text or in the SM (any equation in the main text has to be defined very explicitly in the main text).
- To evaluate the value of this implementation on the proposed application, I would appreciate a comparison of the speed of computation to the sampling rates of data collection and intervention delivery.
- I would have appreciated results experimenting on real data and comparing neural dynamics computed via this method compared to other methods

**Questions:**

NA

---

> ### Author Rebuttal · Authors · 2023-08-08
>
> We thank the Reviewer for their careful reading of our manuscript. The suggestion to better tie our results back to Figure 1 is an excellent one, which we implemented. You can find below our detailed, point-by-point responses to each question raised.
>
> - any equation in the main text has to be defined very explicitly in the main text
>
> We thank the Reviewer for this comment. The Reviewer is absolutely correct and we apologize for the inconvenience. To address this, we refined the presentation of our manuscript. We ensured that every math variable is defined in the main text, obviating the need to navigate to the supplementary material.
>
> - To evaluate the value of this implementation on the proposed application, I would appreciate a comparison of the speed of computation to the sampling rates of data collection and intervention delivery.
>
> Excellent point. Indeed, we could have done a better job tying our results back to the experimental paradigm introduced in Fig. 1. Specifically, we should have discussed how our results advance experimental neuroscience research in practice and can be readily applied to real-time interventional experiments. To address this aspect, we added the following paragraphs to the discussion section:
>
> “To place in perspective the experimental paradigm enabled by CORNN introduced in this work, we now return to the experimental scenario in Figure 1. Imagine an experiment, where mice perform a predefined task several times, say for an half hour, with 3000-4000 observed neurons. This yields roughly ten million parameters to be trained in the dRNN. The current study showcased CORNN's efficacy in training networks of thousands of neurons, requiring just O(10) iterations, each comparable in complexity to gradient computation and taking seconds. Consequently, training such a network from the initial imaging session would take less than a minute. Once trained, the network can enable real-time planning of experimental interventions, testing multiple scenarios in parallel to identify several optimal targeting strategies, a template of potential interventions, for neuron combinations driving desired animal behavior. Moreover, with CORNN's rapid convergence and typical inter-trial intervals of a few seconds in behavioral experiments, the network and the interventional strategy can be refined between trials with incoming data streams, facilitating real-time learning.
>
> To understand the timescales for real-time inference with a dRNN for within-trial intervention experiments using calcium imaging, we can look at typical values. Acquiring each image frame takes about 30 ms (Kim and Schnitzer, 2022). While new frames continue to be acquired, motion correction and neural activity extraction can happen in \sim 5 ms per frame (Zhang et al., 2018). In addition to these ongoing imaging steps, the simulation and decision-making with the CORNN-fitted dRNN is performed. The dRNN starts with the current observed activity, and can quickly simulate multiple future responses under different input scenarios, i.e., potential interventions. Since the simulation involves only iterative matrix multiplications and point-wise nonlinearities, it finishes very quickly - in just a few ms. For instance running a 1000 neuron RNN forward for 10 timesteps (~ a second), for 100 different initial conditions, takes <6 ms on GPU, and <25 ms on CPU, in our hands. Once a desired intervention is identified from the simulations, the phase mask on the spatial light modulator can be updated in \sim 10 ms to stimulate the chosen neurons (Zhang et al., 2018). Putting all the steps together, in under 100 ms, one could capture brain activity, simulate future responses, decide on an intervention, and update the optical stimulation - fast enough for real-time closed-loop application. By streamlining network training, CORNN enables dRNNs for these types of experiments.”
>
> - I would have appreciated results experimenting on real data (...)
>
> Thank you for this comment. Please see our response to the reviewer 1Kip on a similar question.

---

> > ### Comment · Reviewer_Gp7K · 2023-08-14
> > **Response to rebuttal**
> >
> > Thank you very much for your comments.
> >
> > For the added discussion, I am wondering why you did not opt to include processing time comparison as a result?
> >
> > For all other comments, I am quite satisfied with the answers and also with the current score.

---

> > > ### Author Response · Authors · 2023-08-14
> > > **Response to reviewer**
> > >
> > > Thank you for your kind response. When preparing the experiments to include in the results section, we mainly focused on how we could address applicability to a diverse range of attractor structures and model mismatches. In hindsight, it would have been a wonderful idea to add these as results. At the moment, we cannot commit to adding more results, due to the priority of ongoing experiments (Figs. N1-3), but we will try to add a figure on this if time and space permit.

---

### Official Review · Reviewer_GD24 · 2023-07-05

**Soundness:** 3 good
**Presentation:** 3 good
**Contribution:** 2 fair
**Rating:** 5
**Confidence:** 3

**Summary:**

The paper proposes a convex loss for training data constrained recurrent neural networks (dRNNs). This is achieved by performing the optimization in the $d_{t,i}$ space instead of $r_{t,i}$ space (as defined in the paper) and using cross-entropy instead of $l_2$ loss. The authors discuss that this choice makes the optimization convex, and using a weighting scheme they show that a pre-computed inverse Hessian (using the data correlation) makes the optimization much faster than original BPTT or FORCE algorithms. Results are shown on synthetic chaotic rate networks under model misspecification as well as noise and subsampling. Better accuracies and faster runtimes suggest that CoRNN has desirable properties compared to baselines.

**Strengths:**

* Developing tools for online experimentation and targeted manipulation of brain circuits is a significant problem. Given the current experimental advances in the neuroscience field these tools are in crucial demand to help us understand geometrical and computational properties of the brain networks. This makes the problem considered by this paper timely and significant.

* Arguably one of the main bottlenecks of this online experimentation is fitting/inference time complexity. This paper takes a step towards speeding up the optimization by convexifying the loss and pre-computing objects to avoid additional computational burden.

* The motivation is clear and the presentation of the paper is logical, understandable, and easy to follow.

**Weaknesses:**

Below I've included a list of questions and concerns. In summary, I believe that the experimental part of the paper can be improved by a large margin. Specifically since the method is pitched as a general purpose optimization framework it's important to present results on more complex synthetic datasets as well as real neuroscience dataset where the performance of the model is well investigated under various parameter changes. In addition, I think the discussion section can be further improved by being more upfront about the limitations of the work and paths for future work.

> 81 extraction, especially with thousands of neurons, requires GPUs, leaving only the CPU for dRNNs.

* I'm a little confused by this repeating statement, can we not use multiple GPUs? Is this a constraint for the specific hardware you're using or a more fundamental constraint?

* Eq. 1: correct me if I'm wrong but most of the theoretical neuroscience work is done on the equation where the nonlinearity is applied pre-summation $\tau \frac{dr_i}{dt} = -r_i + \sum w_{ij} \phi(r_j)$. Although from a model capacity perspective, the two models are shown to have similar capacities the one I included here is more biologically relevant. Can this be implemented in your approach and can you include some results on this alternative model?

* Related to this, the framework seems to heavily rely on the specific instantiation of the network dynamics. For example, given a different form, we cannot immediately consider the loss form of $d_{t,i}$ and use its linear+sigmoid form. This seems to be an important limitation and some discussion on how to extend this to other forms of ODE would be helpful.


> 114 $d_{t,i} = \tanh(\hat{z})$

* Should it be $\hat{z}_{t,i}$?


* Why is (5) convex with respect to $\theta$? Is this a trivial fact that the authors leave it unexplained?

* Equation 7 is playing a crucial role in the optimization, can the authors bring the proof to the main text instead of SM?

* Based on (S4), we can in principle use other nonlinearities and a corresponding weighting function $c_{t,i}$ right?


**Figures**

* Fig. 1 seems to be over-promising. Although in principle dRNNs can lead to the applications that are discussed but those are not shown in the current work. Unless those applications are not shown in real data/simulations I recommend rewording/having a more concise Fig. 1 that focuses on the contributions. Usually, Fig. 1 is a schematic of what's done in the paper, not an idealistic scenario that can be done in future studies.

* Fig. 3: The colors are very hard to see, specifically because the error bars are too small, can the authors change the formatting to make it easier to read? After reading this paper two times I'm still unsure what "teacher enforcing" means in this context and how it's achieved in various percentages. Instead of these two parameter configurations, can the authors show these as line plots when changing the parameters more continuously on a certain range that includes the performance extremes? Related to this, are the claims true about this specific instantiation of the network or does it hold if we change the noise model? For example, intuitively, cross-entropy loss corresponds to the log probability of a categorical distribution, and as the authors show it provides a better surrogate for the data generated from normal distribution. What if we change the noise model to something like a Poisson or some noise model that's not independent across different time points?

* Does the result hold if we change the dynamical regime of the network? In neuroscience, previous work has hypothesized various network mechanisms such as limit cycles, multi-stability, input-driven networks, transitions between multiple attractors, etc. Is it possible to show results on these other regimes to ensure that the presented optimization framework is not beneficial for a particular regime? Can the authors include experiments with varying the dimension of the chaotic attractor or a more complex data-generating model as opposed to a simple chaotic (low-d) rate network?

* Fig. 4B: The result is a little surprising for me, how can the accuracy go down for other methods? Aren't all of these methods guaranteed to converge to a local minimum? Are the authors controlling for the initialization of the parameters?

* Are the times reported in Fig. 4 CPU times or GPU times?

* Fig. 5: In (A) there are these weird bumps/spikes in the network output. It looks like the errors are happening in a space that's not within the null space of the output. Whereas my experience with the flip-flop task is that BPTT errors don't look weird like this. Is this due to the specific architecture chosen or is this a result of the convex loss/optimization framework?

* Fig. 7: miss-matched -> mismatched

* I'm again surprised that the network can account for the common inputs/unobserved nodes. I think it's an established fact that by just using observational data no method can generically recover the weights in the presence of common inputs and unobserved nodes. This might be the result of the specific data-generating model that's used here. A discussion on this would be helpful to point to the limitations of the work.

* Although the work is motivated by neuroscience data and experiments, no experimental data is included in the paper and that to me is the biggest weakness of the paper. Brain data is much more complex and heterogenous and the noise model, nonlinearity, dynamical regimes, and other factors make it harder to build a robust estimator that generalizes to test trials/test time points. I would assume that various loss functions have different properties that make them more suitable to specific datasets and it's hard for me to believe that a single loss can improve loss on all datasets. Unless results on more complex datasets and simulations are shown it's hard to decide whether the presented framework is preferred to the traditional ones.

**Questions:**

* How are the $d_t$'s initialized to be close to the true ones?

* Since the Hessian is not the exact Hessian, does the optimization still guaranteed to find the (local) minimum? Or is it biased? Can the authors use debiasing strategies to account for the Hessian mismatch?

> 128 $\mathcal{L}_{\text{CoRNN}} = \sum_i \mathcal{L}_i(\hat{\theta}_i)$.

* Is this a trivial fact? Is $\hat{\theta}_{:,i}$ a specific row/column of the weight matrices? This basically means the optimization problem is separable over different rows/columns. This intuitively sounds unrealistic to me unless again I'm missing something important here.

> 111 [...] This turns the potentially infinite time problem into a single time-step one.

* How is this done? The parameters of the model are still $\theta$. If the loss sums over all $t,i$ then the gradients with respect to $\theta$ are still computed in a similar fashion to before unless I'm missing something important here.

> 122 $c_{t,i}=[1-d_{t,i}^2]^{-1}$.

* How is this chosen? This means that some neurons/time points are contributing more to the loss and they are more important to be captured properly, right? Why is this a good choice as opposed to treating all neurons/time points the same?

**Limitations:**

See weaknesses section

---

> ### Author Rebuttal · Authors · 2023-08-08
>
> We appreciate the time the Reviewer put into this detailed review. As they noted, developing tools for real-time experiments and targeted intervention of neural activity is a significant and timely problem. The reviewer also commented that the motivation was clear, logical, and easy to follow. In our revisions, we maintained these strengths while addressing the concerns raised regarding additional experiments and discussion of limitations. We believe the feedback provided has improved the quality of the manuscript substantially. Below we describe the specific changes we made.
>
> - (Request for real data applications and complex synthetic datasets, discussion of limitations)
>
> The reviewer raised important points regarding applying CORNN to real data and testing on more complex datasets. In the revised manuscript, we newly added three more experiments further testing more complex noise conditions in addition to the existing benchmarks. In particular, correlated noise, in both the space and time dimensions, and poisson noise (Figs N1-N3). While application to alternate models like LSTMs and real data remains for future work, the manuscript does include a number of non-random networks as synthetic benchmarks, in Figs. 5 and 7.
>
> In addition, we have expanded the discussion of limitations and future directions in the manuscript, including considerations like nonlinearity mismatch and applications to real data. Please also see our general comments to all Reviewers above about our proposed changes to the abstract. Additionally, in our response to Reviewer 1Kip, we described limitations regarding real data applications and plans to address those in future work.
>
> - (Clarification on CPU usage)
>
> Please see our response to a similar question by the Reviewer 1Kip.
>
> - (Question  on Eq. (1))
>
> Please see our response to a similar question by the Reviewer mPZo.
>
> - (Proof of convexity)
>
> We addressed this comment by making the following change before Eq. (5):
>
> “​​(...) As a result, we replace the L2 loss function with a cross-entropy loss function that is well known to be convex under this estimator. In fact, in the limit of $d_{t,i} \to \pm 1$, the problem reduces to logistic regression. The reader can also verify that the Hessian of this loss, derived in Eq. (S4), is positive semi-definite…”
>
> - Fig. 1 seems to be over-promising. (...)
>
> We made substantial changes to clarify the contribution of our work and how Fig. 1 directly relates to our findings, please see our general comments to all reviewers. Please also see our response to the reviewer Gp7K’s second question.
>
> - (Clarification on teacher forcing)
>
> The teacher forcing in this scenario means that the time activities at time $t$ is computed via Eq. (2), but we replace \hat x_{t,i} in \hat z_{t,i} = \sum_j \theta_{t,j} \hat x_{t,j} with the given data x_{t,i}. \hat x_{t,i} depends on \theta implicitly, so this replacement cuts off the time-dependent computation graph for the gradient. The probability of teacher forcing is the probability with which the replacement is performed at every single time point during the forward propagation of the network. While we currently discuss the approach in supplementary material section S2 and provide code, we will add further implementation detail if this work is accepted. For now, due to time constraints, we focussed on performing additional experiments requested by the reviewers.
>
> - Are the claims true about this specific instantiation of the network or does it hold if we change the noise model?
>
> Please see our general comments regarding the newly added figures.
>
> - Does the result hold if we change the dynamical regime of the network?
>
> Yes, the results hold when networks have specific attractor structures. Please see Figs. 5 and 7 and our proposed changes to the abstract.
>
> - (Surprising Fig. 4B results)
>
> The accuracy in these graphs are measured through the correlations between learned and original networks. The models are minimizing errors defined on the predictions. It is not possible to define a loss function on the ground truth weights, nor is it experimentally relevant.
>
> - (Unexpected spikes in Fig 5A)
>
> To clarify, Fig 5 A is an illustrative example of a network that arises when the specific architecture we describe in Eq. (1) is trained with BPTT, not CORNN, to perform a 3-bit flip flop task (See also the spikes in Fig. 2 of Sussillo and Barak, 2013). Therefore, this figure or any error the network makes has no relationship with CORNN or the convex loss. Based on the reviewer’s feedback, we have updated lines 217-218 to explicitly state that we used BPTT for training networks to perform the 3-bit flip flop task in Fig. 5(A).
>
> - (observational data vs synaptic connectivity)
>
> We are not interested in reproducing synaptic connectivity. We only used it as a surrogate to test the reproduction accuracy in cases where generator and inference networks followed the same dynamical system equations. To clear any confusion regarding the learned connectivity matrix, we added a paragraph to the discussion, though omitted here due to space limitations.
>
> - (no real data + how to decide if our work is preferred to traditional ones)
>
> Please see our general response, as well as response to Reviewer 1Kip for the changes we made to address this comment.
>
> - (d_t initialization)
>
> They can be initialized using theta_ls as defined in Eq. (11). We added the following sentence right after Eq. (11) to clarify this:
> “Then, the initial set of predictions becomes $\hat d_{t,i} := \tanh(\sum_j x_{t,i} \theta_{\rm ls})$, which is subsequently refined through the HAPE iterations.”
>
> - (Inexact Hessian and convergence)
>
> Since the approximate Hessian is positive semi-definite, the parameter update direction is guaranteed to be a descent direction. We updated the relevant part in supplementary text.
>
> - Question on lines 128, 111 and 122
>
> We noted the relevant supplementary sections in the main text so that the reader can find them easily.

---

> > ### Comment · Reviewer_GD24 · 2023-08-12
> >
> > I thank the authors for responding to some of my questions and incorporating some of my comments. For the ones you've already answered:
> >
> > - **Request for real data applications:** I still think that not including any experimental result is a big weakness of the paper. I read the global responses and replies to other reviewers. The main motivation of the paper is to develop these tools for fast real-time inference in experimental data. While I think the developed methods are quite interesting, I think the type of model mismatches, nonlinearities, dynamical regimes, and noise in real data is quite distinct from the ones considered in the experiments. That said, given that the machinery for developing the proposed tools is relatively elementary, without showing results on real biological datasets it's hard to recommend acceptance for this paper in NeurIPS. Given the availability of public neural datasets, I don't think it's a big request to report results on some of these existing datasets with simple APIs to access them.
> >
> > - **Clarification on CPU usage, Question on Eq. 1, Fig. 1 seems to be over-promising, Clarification on teacher forcing, Specific instantiation of the network, Fig. 4B results:** I thank the authors for clarifying these.
> >
> > - **Observational data vs synaptic connectivity:** I believe this question is similar to limitation 1 mentioned by review *mPZo*. While it's replied that the methods are trivially extensible to other cases I'd appreciate some math showcasing that in the discussion here.
> >
> >
> > Here's a list of the questions that remain unanswered:
> >
> > **I'm again surprised that the network can account for the common inputs/unobserved nodes.**
> > Although the experiments show that training models using CORNN allows for the recovery of the connectivity matrix I'm still having a hard time comprehending this result. There are so many non-identifiability results in the statistics and neuroscience literature showing that even with infinite observations and convex loss the true parameters are still not recoverable because they suffer from identifiability, meaning that there are classes of the parameters that produce the exact same observations. This is exacerbated by considering unobserved nodes and common inputs. It shouldn't be hard to come up with examples where the recovered matrices are arbitrarily far from the true ones while the loss is small because the generated signals are close to the true ones. There is very little discussion on this in the paper. I think a more thorough empirical assessment is needed to find the regimes in which the proposed model is not able to recover the true matrix.
> >
> >
> > **Why is (5) convex with respect to? Is this a trivial fact that the authors leave unexplained?**
> >
> >
> > **128: Loss is separable for different columns of $\theta$: Is this a trivial fact? Is _ a specific row/column of the weight matrices? This basically means the optimization problem is separable over different rows/columns. This intuitively sounds unrealistic to me unless again I'm missing something important here.**
> >
> >
> > **Choice of $c_{t,i}$: How is this chosen? This means that some neurons/time points are contributing more to the loss and they are more important to be captured properly, right? Why is this a good choice as opposed to treating all neurons/time points the same?**

---

> > > ### Author Response · Authors · 2023-08-12
> > > **Response to technical questions**
> > >
> > > We thank the reviewer for their willingness to discuss further and provide valuable feedback. At this point, it seems natural to divide the topics of discussion into two: technical aspects regarding convex optimization and philosophical concerns regarding neuroscience applications. For a productive discussion, we will first answer the questions on the former.
> > >
> > > - Why is (5) convex with respect to theta?
> > >
> > > The convexity of the loss function comes from two facts:
> > >
> > > 1) Linear + sigmoid structure is well known to be convex under cross-entropy. The resulting Hessian for this particular case is $ H_{kl} = \sum_t (1-\hat d_t^2) x_{t,k} x_{t,l} + \lambda \delta_{kl}$, which is the Hessian of the logistic regression. We can prove that this Hessian is positive semi-definite by showing: $z^T H z = \sum_t (1-\hat d_t^2) (\sum_k  x_{t,k} z_k)^2 + \lambda \sum z_k^2 \geq 0$ since $(1-\hat d_t^2)\geq 0$.
> > >
> > > 2) We take a non-negative weighted sum of this loss, and non-negative weighted sum of convex functions preserves convexity.
> > >
> > > - (...) I'd appreciate some math showcasing (application to leaky-current RNNs) in the discussion here.
> > >
> > > In the leaky-current RNNs, the dynamical system equations follow Eq. (S27). Specifically, we can write the teacher-forced estimator as:
> > >
> > > $\hat r(t) = \tanh\left[ (1-\alpha) \text{arctanh}(r(t-1))  + \alpha W_{\rm in} u(t) + \alpha W_{\rm rec} r(t-1)   \right]$.
> > >
> > > Here, due to teacher-forcing, $r(t-1)$ and $u(t)$ are defined by the data. Hence, the estimator simply has the form: $\hat r(t) = \tanh( \theta x(t) )$, where $\theta$ is a set of concatenated weight matrices and a non-learnable bias term, which can be enforced as an equality constraint or just left untrainable, whereas $x(t)$ is concatenated input (u(t)), previous time activations (r(t-1)), and a vector of 1s that multiply the bias. Now, we simply can exploit the fact that r_{t,i} follows a linear + tanh structure, hence (1+r)/2 will be a linear + sigmoid, in the leaky-current RNNs and replace \hat d_{t,i} with \hat r_{t,i} wherever applicable in our original calculations.
> > >
> > > - 128: Seperability of loss
> > >
> > > The separability of the loss function was shown in Eq. (S1) in supplementary materials. With the choice of the loss function, and the teacher-forcing, the loss function becomes separable across the neurons and hence leads to independent problems across the columns of theta. To clarify, we updated the text after Eq. (S1) as follows:
> > >
> > > “We note that for a given $i$, the functions $\mathcal L_i$ depend only on $\hat \beta^{(i)}: = \hat \theta_{:,i}$, e.g. the $i$th column of the $\hat \theta$ matrix. In other words, the lack of cross-talk between the columns of the weight matrix in the loss function means each output can be regressed independently with respect to the corresponding part of the loss function, \emph{i.e.}, $\mathcal L_i(\hat \theta_i)$.”
> > >
> > > - Choice of c_t
> > >
> > > The main motivation driving the choice of c_t was to turn the Hessian into a pre-computable quantity that was aligned across all subproblems. This is our unique contribution and is explained right before Eq. (4).
> > >
> > > That being said, this particular choice was motivated from a theoretical point of view as well. We discussed this in the supplementary material Section S1.3. Please note that this choice of c_t does not mean that certain neurons are contributing more to the L2 error. In fact, in the low error limit, we showed in Section S1.3 that minimizing the CORNN loss is approximately equal to minimizing the L2 error on the currents (z(t)), where each data-point contributes equally to the loss.
> > >
> > > - “given that the machinery for developing the proposed tools is relatively elementary.”
> > >
> > > We believe the main misunderstanding between our view of our work and the reviewer’s is well summarized in this sentence by the reviewer. Please allow us to elaborate.
> > >
> > > The choice of c_t, and the subsequent alignment of subproblem Hessians, are both novel advancements that allow this rapid increase in the training speed. For example, see the traditional exact Hessian updates in Fig. 4. For networks with hundreds of neurons, they are 1 order of magnitude slower compared to CORNN. For larger networks in Fig. N2, it is not reasonable to compute Hessian multiple times at each optimization step.  In this scenario, gradient on the convex loss (with ADAM) trains in roughly an hour (experiment ongoing) vs the seconds it takes for CORNN. Hence, the solver we are developing here is unique and novel even when looked through the lens of minimizing a large convex problem.
> > >
> > > Thank you for your time and please let us know if you have any additional questions on the convex optimization aspects. Once we are aligned in these aspects, we are confident that we will be able to reach a mutual agreement on the neuroscience aspects as well.

---

> > > > ### Comment · Reviewer_GD24 · 2023-08-14
> > > >
> > > > I thank the authors for addressing my technical questions. I wish to clarify that my following statement "given that the machinery for developing the proposed tools is relatively elementary" did not stem from misunderstanding the contributions. Rather, while it's clear what the proposed contributions are, my questions were mainly to ensure the validity of the claims.
> > > >
> > > > Indeed I do find the ideas attractive for the neuroscience community, however, I believe that in order to prove the applicability of these ideas to neuroscience experiments, at least some proof of concept results need to be accompanied. As I understand, the details of the developed method rely heavily on the form of the model equation. Therefore, the reparameterization and convexification must be re-instantiated for small changes in the underlying model. Given that, it is important to show that the current model with no changes works well on some neuroscientific datasets.
> > > >
> > > > While I still look forward to discussing the other points I brought up (1. identifiability and the recovery of the true matrix in the presence of unobserved nodes, and 2. results on real data) I encourage the authors to update the manuscript with the details included in response to my and other reviewers' questions (1. leaky-current RNN, 2. a clearer proof of convexity and separability, 3. the reasons behind the choice of $c_t$, etc.). Some of these points are already included in the paper, but it can be made clearer to improve the presentation of these ideas.

---

> > > > > ### Author Response · Authors · 2023-08-15
> > > > > **Response to remaining points - part I**
> > > > >
> > > > > Thank you for the quick response and continued feedback. We will be refining the presentation of our manuscript by keeping these three points in mind. Please find attached our detailed response to the neuroscience related concerns raised by the reviewer.
> > > > >
> > > > > - Request for real data applications
> > > > >
> > > > > The reviewer holds the position that if we take publicly available data and do a train-test split, we will be able to test the real-data applicability of CORNN. We disagree with this premise on following grounds:
> > > > >
> > > > > 1. Neural activities during trials are stereotypical. Hence, even when trials are split into train and test sets, there will be a significant leakage. The fact that the CORNN-fitted dRNN can provide a low error in the test set does not provide any real quantification of its ability as a generative model, but only shows that one can achieve low training error.
> > > > >
> > > > > 2. dRNNs have millions of parameters. In contrast, observed neural activities tend to lie on very low dimensional manifolds, as low as less than 10 (See Rumyantsev et al. 2020 and Valente et al. 2022). In this regime, almost any network can fit the observed neural activity and make reasonable predictions.
> > > > >
> > > > > Due to these reasons, we believe that simply showing a low test score in a neural dataset is not a true test of real-data applicability, nor should it be taken as such by the community. For example, what would it mean to have a low score? How low is good enough? There is simply no ground truth, and surrogate tests with observational data are, in our opinion, unreliable.
> > > > >
> > > > > The true test of dRNNs’ applicability to real-data would require interventional experiments. In this test, the network learned by the CORNN would be tested under novel perturbed conditions, behaviorally or neurally, and not through a training-test split. We emulated this in Figs. 7 and 8 on a synthetic benchmark, but an experimental counterpart is left as future work. In summary, applicability to real data should not be addressed with a simple figure created from a random publicly available data. For this reason we solely focused on synthetic benchmarks, where we know the ground truth and can test whether dRNNs can reproduce the underlying attractor structures. Please see the works we cited in line 55, i.e. [22-27], for previous application of similar networks to real-data. We will add a paragraph to the discussion explaining these aspects for any potential reader with similar concerns as the reviewer.
> > > > >
> > > > > - Questions on synaptic matrix recovery and non-identifiability issues
> > > > >
> > > > > We thank the reviewer for this comment, as it brings up an important point. As the reviewer points out, it is well known that in general, neural networks are non-identifiable, and that the input-output function a neural network implements is degenerate in the weights. Given this fact, to what extent should we expect our CORNN trained dRNNs to recapitulate the generator synaptic weights?
> > > > >
> > > > > Although the answer depends on the specifics of the setup, a full discussion of this issue is beyond the scope of our paper, and also besides the point. We did not mean to suggest that recovering the generator synaptic connectivity matrix is the gold standard of what CORNN aims to do. We state in the paper that “Despite observing only 500 out of 5000 neurons and experiencing an average 10% miss-match between $α_G$ and $α_I$ , the CoRNN-learned network successfully reproduced both the neural dynamics and the output.” In fact, in Figs. 7 and 8, we show that the correlation between generator and dRNN synaptic weight matrix is quite suboptimal (r=0.45, Fig. 7D), even in the best case with 100% observation of the generator dynamics.
> > > > >
> > > > > Despite this fact, CORNN is able to match the neural dynamics and the output in a novel perturbation trial, a finding that is quite in line with the fact that neural nets are non-identifiable. This suggests that, at least in the context of stereotypical trial activities, the state-space structure is captured by CORNN even though the synaptic connectivity is not. We note that the real test here was whether CORNN-fitted dRNNs could reproduce the neural activities under novel perturbations, a test we wouldn’t be able to perform with observational neural data.

---

> > > > > > ### Author Response · Authors · 2023-08-15
> > > > > > **Response part II**
> > > > > >
> > > > > > All that being said, it is clear that our writeup could be a lot more clear about this issue, and have added the following changes to the result section:
> > > > > >
> > > > > > “When we subsampled the generator networks by 10% and introduced jitter in time scales, we observed that the inference model reproduced the neural dynamics in the observed population in a novel perturbation trial (See Fig. 7C), even though the sub-connectivity matrix was not well reproduced (See Fig. 7D-E). This reinforces the findings of previous literature that the connectivity matrices learned by dRNNs should be interpreted as functional connections, accounting for the behavior of networks at the level of neural dynamics, and not synaptic connections [Perich et al 2021, Das et al 2020].”
> > > > > >
> > > > > > We have additionally added this paragraph to the discussion:
> > > > > >
> > > > > > “An important point in considering the use of CORNN in experimental settings comes from the fact that neural networks are, in general, non-identifiable [citation]. That is, for any given settings of the parameters, there are other settings which give the same input-output function. This means that CORNN should not be used to try to infer the true underlying synaptic connectivity matrix from a dataset. Instead, the main use of CORNN is to infer an RNN model which recapitulates the dynamical trajectories in a neural population. It is further possible that CORNN can capture the underlying attractor structure of a system. However, we caution that any claim having to do with attractor structures must be experimentally validated with perturbation experiments that directly test attractors. In our work, we simulated such experimental validation (Figs. 7 and 8), and found that in the setting tested, CORNN was indeed able to predict the dynamical effects of perturbations on the neural population.”

---

> > > > > > ### Comment · Reviewer_GD24 · 2023-08-16
> > > > > >
> > > > > > I thank the authors for further continuing the discussion.
> > > > > >
> > > > > > **Identifiability:** Indeed the low correlation in the partial observation regime (while attaining high accuracy in neural firing rate generation) is a signature of non-identifiability, even when the data is generated from the true model. The paper in its current form gives the impression that due to the convexity of the loss the method is able to recover the weights exactly, hence solving the identifiability issue, which cannot be true. Given the clarification, I would encourage the authors to include a short discussion in the revised manuscript and be upfront about what can or cannot be achieved by the method.
> > > > > >
> > > > > > **Experimental Data:** I respectfully tend to disagree with this argument for a few reasons:
> > > > > > - There are many public datasets that are not trial-based. For the cases where the data is a continuous time series achieving a low test error (where the model that's trained using time points 1 to $T$ predicts time points $T+1$ to $T+T_{test}$) has been a fundamental challenge in computational neuroscience and I'm not aware of any methods with excellent performance.
> > > > > > - Indeed a better test of the model is on interventional data, but many labs are still not able to build the experimental apparatus to incorporate such experiments, while still interested in applying these models to their observational data with less costly computational resources.
> > > > > > - Regarding how low is good enough, there are statistical tests that can address this but just naively, one would expect to see comparable-if not better-results than the existing methods. Additionally, one can investigate further and establish neural or behavioral relevance of the fitted weights in a more exploratory setting (this is not well-thought, but just as an example, if the authors can show that CORNN's weights are closer to Dale's law as opposed to BPTT).
> > > > > > - Although low-dimensional structures are prevalent in neuroscience experimental data, the dynamics are still complex and may require many parameters to be explained. In addition, the overparameterization of these models can help achieve a better test error.
> > > > > >
> > > > > >
> > > > > > I'm going to increase my score due to the excellent discussion points and clarifications, but I still stand by my argument that the lack of results on experimental data is a weakness of the paper, and if accepted, I look forward to seeing future iterations of this work with applications in real experimental settings.

---

> > > > > > > ### Comment · Reviewer_GD24 · 2023-08-16
> > > > > > >
> > > > > > > Also, please replace all instances of "miss-match" with "mismatch", I believe the former is wrong.

---

> > > > > > > > ### Author Response · Authors · 2023-08-16
> > > > > > > >
> > > > > > > > We thank the reviewer for their willingness to discuss further and for helping us improve our work. We will add the discussion on the non-identifiability and fix the typos with "mismatch." We appreciate your inputs regarding the real-data applications, which we will keep in mind moving forward.
> > > > > > > >
> > > > > > > > Thank you for your comments and questions throughout the entire review process. The manuscript has been made much better by them.

---

### Official Review · Reviewer_1Kip · 2023-07-07

**Soundness:** 4 excellent
**Presentation:** 3 good
**Contribution:** 4 excellent
**Rating:** 8
**Confidence:** 4

**Summary:**

The authors introduce a method to produce a digital twin of a biological neural network that can be inferred from measurements of neuronal activity.  The major advantage of the model is that it can be trained in real time to gain the basic dynamics of neural population. In the future this could allow advance HRI applications, like linking brain activity to controlling prosthesis.

The digital twin is trained using a randomised neural network that represents a neural population. It is used as a teacher network for student network that has a specific structure, designed in a way this is allows approximate convexification of the optimisation problem, enabling a ridiculously easy training of the network that can scale to large neural populations.  The computational cost of performing the analysis of the biological neural network scales linearly with its size.

The system is evaluated by a test, where a random neural network is initialised repeatedly with Gaussian random weights in a range that can produce also chaotic self-sustained signalling. The is assumed to provide a large enough variation for a neural signal generator


**Strengths:**

The real time, fast learning capability of the suggested architecture. The convexificaiton, as a technology. has definitely uses and application potential beyond the domain of the paper.

Excellent figures to clarify the process.


**Weaknesses:**

The reasoning about the enforced use of CPU does not sound convincing to me. See the question part.

**Questions:**

on line 81: Why would the GPU use of real time acquisition of thousands of cell signals would force the dRNN calculation to CPUs? As these are tasks that can be pipelined,  the obvious answer is to use more GPUs .  Also, on can expect that signal detection part will be moved to special purpose chips tuned for inference as the signal dimension grows.

There has been a recent article (https://doi.org/10.1038/s41598-022-25421-w, published Jan 2023 ) on data-driven modelling of C. elegans brain with different recurrent neural networks architectures that found that 4-hidden layer GRU is able to accurately reproduce the stimuli. As this is based on comparing measured data to artificial neural network behaviour like in the current manuscript, without the real time acceleration, but using  back propagation training, it would be a very good test case for the advantages of using the CoRNN.
Perhaps the Authors could add this to their submission.

---

> ### Author Rebuttal · Authors · 2023-08-08
>
> We thank the Reviewer for their comments and endorsement of our work. We carefully reviewed each comment and have provided below our responses detailing how we changed the manuscript to incorporate the Referees’ suggestions. We would also greatly appreciate it if the Reviewer would like to propose any additional research areas upon which  the specific convexification trick we developed might have an impact, for potential mention in the Discussion.
>
> - Why would the GPU use of real time acquisition of thousands of cell signals would force the dRNN calculation to CPUs?
>
> We thank the reviewer for this comment. The reviewer is totally correct - distinct analyses and algorithms can run in parallel on different GPUs if the computer has multiple GPUs, or analysis can be put into FPGAs (for example see https://elifesciences.org/articles/78344), or other computational resources. In general, it is the case that at least one GPU will be unavailable during an online neuroscience imaging experiment for things like online motion correction and cell extraction, and so we wanted an algorithm that works well with limited computational resources. That being said, given that a typical contemporary neuroscience experiment is offline, we designed CORNN to benefit from the availability of GPUs, as shown in Fig. S2, S4, and S5. We have changed the manuscript with these issues in mind in the following ways.
>
> First, we showed that in a large-scale scenario, GPU accelerated gradient descent on Pytorch falls behind CORNN by 1-2 orders of magnitude (experiment still running, see Fig. N2 in the appended PDF for the current stage of the experiment). Hence, CORNN is valuable even when GPUs are available. We plan to replace current Figure 4, which will become a supplementary figure, with this experiment if our work is accepted for publication.
>
> Second, we revised lines 79-82 of the manuscript to clearly state that GPUs may be available for the real-time training as well and incorporated the discussion above regarding the on-chip processing:
>
> “Motivated by the goals described above, our paper focuses on training dRNNs accurately and as fast as possible. However, experimental concerns and the need for biological interpretability lead to several constraints. Firstly, to ensure real-time communication with the experimental apparatus, we require that the training process takes place on standard lab computers, not clusters. Secondly, given the computational complexity of real-time processing in large scale recordings, especially with thousands of neurons, we expect that at least one of the GPUs is reserved for extracting neural activities from brain-imaging movies (though see Chen et al. for a promising development in smaller scale experiments), or spikes from electrophysiological recordings, leaving the central processing unit (CPU), or perhaps a second GPU, for training dRNNs…”
>
> - There has been a recent article on data-driven modelling of C. elegans brain (...). (Request by the reviewer for real data application)
>
> We thank the reviewer for bringing this relevant work to our attention. We now cited this work in our manuscript, where we discuss data-constrained applications of neural networks (line 39).
>
> We perfectly agree that real-data applications would be very helpful to illustrate the usefulness of our approach. However, as it stands, the manuscript is 11 pages long after revisions, which we need to trim if accepted. The previous work (see line 55) aimed to apply existing optimization methods to train these dRNNs on real-data and already demonstrated the validity of this paradigm, though via an inefficient training paradigm that takes days to converge in our new experiment (Fig. N2).
>
> We wanted our work to convey one powerful idea and in an accessible manner for readers from multiple fields. Our work is a necessary step towards the development of dRNNs as an experimental technology, especially for real-time interventions, but further steps should be taken. To address this, we added the following paragraph to the discussion:
>
> “All in all, this work constitutes a first step towards the application of CORNN to experimental data. However, several steps remain to apply dRNNs in real-time to interventional experiments. Some example steps may include transformation of calcium traces, or electrophysiological spikes, into firing rates normalized within $[-1,1]$, applying CORNN solver developed in this work into first offline than online experimental scenarios, estimation of neuronal time-scales from the experimental data instead of tuning them as hyperparameters (See Table 2 in \cite{perich2021inferring} for tuned hyperparameters) and perhaps developing a low-rank regularization approach into CORNN that opens the door to interpreting observed dynamics in terms of latent variables \cite{beiran2021shaping,valente2022extracting}.“
>
> To some extent, that previous work applying dRNN training to real data necessitated the fine-tuning of several hyperparameters [See Table 2 in \cite{perich2021inferring} for tuned hyperparameters] constitutes evidence that applications to real-data are non-trivial and will require further work. For now, we address these issues as limitations in the Discussion section, and we lay out a feasible path for future research.

---

> > ### Comment · Reviewer_1Kip · 2023-08-15
> >
> > Than you for providing clarification and addressing my concerns.
> >
> > Having a form of the neural network that is convex and general enough to address complicated time series could also be valuable in building maintenance systems - to distinguish indicators for potential failures, or in situation that require fast reaction times like cyber attacks in communication networks. Of course, you have to adapt your idea accordingly.

---

> > > ### Author Response · Authors · 2023-08-16
> > >
> > > Thank you for the comments and for your help on this manuscript! This line of research seems interesting, and though it is outside of our area of expertise we will look into it.

---

### Author Rebuttal · Authors · 2023-08-09

We thank the AC for considering our work and all reviewers for their expert reports, which we used to refine the presentation of our manuscript and add new experiments. We note that two reviewers thought the paper had excellent impact in its area and one thought there were no major concerns. The comments from the fourth reviewer made us realize that the presentation of the manuscript needed to be improved. We acted on all suggestions by making appropriate changes in the manuscript as detailed below.

We are sharing a PDF file with three new figures (Figs. N1-3) as part of the revision. Please note that we have fewer samples than desired in the current version of these figures due to time limitation. If our work is accepted, we will increase the number of samples in all figures. The figure descriptions are provided within the PDF. Below, we list the points we aimed to address with each new figure:

- Fig. N1: A new experiment with correlated noise, for the networks considered in Fig. 7. As requested by Reviewers mPZo and GD24, we wanted to test whether correlated noise would prevent CORNN from training. Though CORNN trains more accurately under comparable independent noise, increasing the number of trials helped mitigating correlated noise.

- Fig. N2: A new experiment with a chaotic network of 5000 neurons. So far, our simulations tested networks with hundreds of neurons due to inefficiency of other algorithms. With this new experiment, we showed how previous algorithms failed to scale.

- Fig. N3: A similar experiment to (old) Figure 4, but with independent Poisson noise. As requested by Reviewer GD24, we tested the accuracy of CORNN under asymmetric non-Gaussian noise distribution.

Moreover, we refined the presentation of the manuscript significantly by incorporating feedback from the reviewers. Specifically, we made the following changes:

- We restructured the Introduction and Discussion to explain clearly how the rapid inference made possible via CORNN enables real-time interventional experiments, as well as the next steps needed to actualize this experimental scenario. We provided detailed descriptions of these changes in our point-by-point response to the Reviewers wherever applicable.

- We updated the Introduction to emphasize why CORNN is novel and timely. Specifically, we re-wrote the following paragraph:

“Despite recent advances in the dRNN framework, how to perform fast and scalable reconstructions of neural activity traces from large-scale empirical recordings has remained unclear. Notably, the slowness of existing dRNN optimization algorithms may often necessitate the use of high-performance clusters and several days of computation in large-scale experiments. These limitations have been a barrier to the widespread adoption of dRNN approaches. To fully harness the potential of  dRNNs, it is essential to extend their range of applicability from offline analyses to on-the-fly applications within individual recording sessions. Therefore, an approach to training dRNNs in a nearly immediate manner would be a key advancement that would enable novel experimental approaches, such as theory-driven real-time interventions targeting individual cells with specific computational or functional roles (Fig. 1). However, realizing these benefits requires having an optimization routine that is fast, robust, and scalable to large networks.”

Finally, inspired by the reviewer reports, we propose to make the following changes to the title and abstract to refine the presentation:

Proposed Title: CORNN: Convex optimization of recurrent neural networks for rapid inference of neural dynamics

Proposed Abstract:
Advances in optical and electrophysiological recording technologies have made it possible to record the dynamics of thousands of neurons, opening up new possibilities for interpreting and controlling large neural populations. A promising way to extract computational principles from these large datasets is to train data-constrained recurrent neural networks (dRNNs). Making this training real-time could open doors for research techniques and medical applications to model and control interventions at single-cell resolution and drive desired behavior. However, existing training algorithms for dRNNs are inefficient and have limited scalability, making it a challenge to analyze large neural recordings even in offline scenarios. To address these issues, we introduce a training method termed Convex Optimization of Recurrent Neural Networks (CORNN). In studies of simulated recordings of hundreds of cells, CORNN attained training speeds ~ 100-fold faster than traditional optimization approaches while maintaining or enhancing modeling accuracy. We further validated CORNN on simulations with thousands of cells that performed simple computations such as those of a 3-bit flip-flop or the execution of a timed response. Finally, we showed that CORNN can robustly reproduce network dynamics and underlying attractor structures despite mismatches between generator and inference models, severe subsampling of observed neurons, or mismatches in neural time-scales. Overall, by training dRNNs with millions of parameters in subminute processing times on a standard computer, CORNN constitutes a first step towards real-time network reproduction constrained on large-scale neural recordings and a powerful computational tool for advancing the understanding of neural computation.

The first change in title is emphasizing that it is optimization, not the RNN, that is convex. The second change is regarding the “unit resolution,” which was unnecessarily restrictive since dRNNs could be applied to non-unit resolution activity. Inspired by the comments of Reviewers GD24 and mPZo, the final change emphasizes what exactly we achieved towards a real-time application, i.e. that our work allows rapid inference. Abstract changes are cosmetic in nature to refine the presentation. We would appreciate reviewers' feedback.

---

### Decision · Program_Chairs · 2023-09-21

**Decision:**

Accept (poster)

**Comment:**

The heart of this work is to provide a convex approximation to RNN learning where the learning is reduced in complexity to the point where dynamical systems can be learned online. The primary points are to 1) replace the full cost with a "step-by-step" cost wherein the dynamics are learned between every two pairs of measurements instead of learning a full sequence that matches the data at all points and 2) replace the standard cost with a cross entropy cost. Given these changes the authors show that the dynamics can be learned efficiently even at scale.

The work has many merits in terms of the goals and methods. The main detriments came in the form of confusion brought on by the terminology and presentation, that the authors were able to respond to in the response period. To add to this discussion from the AC point, the "teacher" terminology is confusing since full-FORCE (DePasquale et al. 2018 PLOS ONE) uses a "teacher" network that serves a distinctly different role that also minimizes instability during learning (this work should also probably be cited). There is also some commentary about vanilla RNNs being interpretable, which is a point still up for debate. I believe with the appropriate clarification changes promised by the authors, this work can be a nice addition to the NeurIPS conference and I thus recommend this work be accepted.